# Impact of gestational low-protein intake on embryonic kidney microRNA expression and in nephron progenitor cells of the male fetus

**Letícia de Barros Sene**[1], **Wellerson Rodrigo Scarano**[2], **Adriana Zapparoli**[1], **José Antônio Rocha Gontijo**[1], **Patrícia Aline Boer**[1]*

**1** Fetal Programming and Hydroelectrolyte Metabolism Laboratory, Nucleus of Medicine and Experimental Surgery, Department of Internal Medicine, Faculty of Medical Sciences at State University of Campinas, Campinas, SP, Brazil, **2** Department of Structural and Functional Biology, Bioscience Institute, São Paulo State University, Botucatu, SP, Brazil

* boer@fcm.unicamp.br

## Abstract

**Data Availability Statement:** Data are available in NCBI: (https://www.ncbi.nlm.nih.gov/sra/PRJNA694197) Further information can be found: (https://bv.fapesp.br/pt/pesquisador/671860/

### Background

Here, we have demonstrated that gestational low-protein (LP) intake offspring present lower birth weight, reduced nephron numbers, renal salt excretion, arterial hypertension, and renal failure development compared to regular protein (NP) intake rats in adulthood. We evaluated the expression of various miRNAs and predicted target genes in the kidney in gestational 17-days LP (DG-17) fetal metanephros to identify molecular pathways involved in the proliferation and differentiation of renal embryonic or fetal cells.

### Methods

Pregnant Wistar rats were classified into two groups based on protein supply during pregnancy: NP (regular protein diet, 17%) or LP diet (6%). *Renal miRNA sequencing (miRNA-Seq)* performed on the MiSeq platform, *RT-qPCR of predicted target genes*, immunohistochemistry, and morphological analysis of 17-DG NP and LP offspring were performed using previously described methods.

### Results

A total of 44 miRNAs, of which 19 were up and 25 downregulated, were identified in 17-DG LP fetuses compared to age-matched NP offspring. We selected 7 miRNAs involved in proliferation, differentiation, and cellular apoptosis. Our findings revealed reduced cell number and Six-2 and c-Myc immunoreactivity in metanephros cap (CM) and ureter bud (UB) in 17-DG LP fetuses. Ki-67 immunoreactivity in CM was 48% lesser in LP compared to age-matched NP fetuses. Conversely, in LP CM and UB, β-catenin was 154%, and 85% increased, respectively. Furthermore, mTOR immunoreactivity was higher in LP CM (139%) and UB (104%) compared to that in NP offspring. TGFβ-1 positive cells in the UB increased by approximately 30% in the LP offspring. Moreover, ZEB1 metanephros-stained cells

leticia-de-barros-sene/) (https://repositorio.unesp.
br/handle/11449/148594).

**Funding:** This work was supported by Fundação de
Amparo à Pesquisa do Estado de São Paulo
(FAPESP, 2013/12486-5 and 2014/50938-8),
Coordenação de Aperfeiçoamento de Pessoal de
Nível Superior (CAPES) and Conselho Nacional de
Desenvolvimento Científico e Tecnológico (CNPq,
465699/2014-6).

**Competing interests:** Competing interest: No
conflicts of interest, financial or otherwise, are
declared by the authors

**Abbreviations:** Bax, Apoptosis regulator; Bcl-2, B-
cell lymphoma 2; Bcl-6, B-cell lymphoma 6; Bim,
or Bcl-2-like protein 11; BSA, Bovine serum
albumin; cDNA, complementary deoxyribonucleic
acid; CEUA/UNESP, Institutional Ethics Committee;
c-Myc, regulator genes and Myc proto-oncogenes;
CM, metanephros cap; c-ret, rearranged during
transfection; DAB, 3,3'- diaminobenzidine
tetrahydrochloride; DNA, deoxyribonucleic acid;
DG, days of gestation; GAPDH, Glyceraldehyde 3-
phosphate dehydrogenase; GDNF, Glial cell line-
derived neurotrophic factor; IGF1, Insulin-like
growth factor 1; Ki-67, nuclear protein associated
with cellular proliferation; let-7, lethal-7 (let-7)
gene; Lin28b, Lin-28 Homolog B, suppressor of
microRNA (miRNA) biogenesis; LP, gestational
low-protein intake; Map2k2, mitogen-activated
protein kinase kinase 2; *miRNA (miR)*, small non-
coding RNA molecule; *miRNA-Seq*, *miRNA
transcriptome sequencing*; MM, metanephric
mesenchyme progenitor cells; mRNA, messenger
ribonucleic acid; mTOR, mammalian target of
rapamycin; mTORC1, mammalian target of
rapamycin complex 1; NGS, Next Generation
Sequencing; NOTCH1, single-pass transmembrane
receptor protein; NP, normal protein intake; PCNA,
Proliferating cell nuclear antigen; PRDM1, PR
domain zinc finger protein 1; RIN, RNA Integrity
Number; RT-qPCR, reverse transcription-
polymerase chain reaction quantitative real-time;
Six-2, SIX homeobox 2; TGFβ-1, transforming
growth factor beta 1; UB, ureter bud; U6 and U87,
internal reference gene; WT1, Wilms' tumor
protein; ZEB 1 and ZEB 2, Zinc finger E-box-binding
homeobox 1 and 2; 17-DG, 17-days LP fetal
kidney.

increased by 30% in the LP offspring. ZEB2 immunofluorescence, although present in the entire metanephros, was similar in both experimental groups.

## Conclusions

Maternal protein restriction changes the expression of miRNAs, mRNAs, and proteins involved in proliferation, differentiation, and apoptosis during renal development. Renal ontogenic dysfunction, caused by maternal protein restriction, promotes reduced reciprocal interaction between CM and UB; consequently, a programmed and expressive decrease in nephron number occurs in the fetus.

## Introduction

The lack of nutrients may result in signaling changes in pivotal pathways during various stages of fetal development, which may cause irreversible organ and system disorders in adulthood [1]. Fetal programming refers to any insult during development, which causes long-term effects on an organism's structure or function [2]. Disruptions in fetal programming result in low birth weight, fewer nephrons, and increased risk of cardiovascular and renal disorders in adulthood [3–6]. Studies by other authors and us have demonstrated lower birth weight, 28% fewer nephrons, reduced renal salt excretion, chronic renal failure, and arterial hypertension in gestational low-protein (LP) intake compared to standard (NP) protein intake offspring in adulthood [3–7].

Nephrogenesis involves tight control of gene expression, protein synthesis, and tissue remodeling. Studies have demonstrated that nephron numbers are determined by the interactions between ureter bud (UB) and metanephros mesenchyme (MM) progenitor cells [8–10]. Signals from MM induce UB stimulated growth and branching of the tubule system. In turn, MM proliferation and differentiation, constituting a mesenchymal cap (CM), is mediated by UB ends [11].

There has been serious interest in the role of epigenetic changes, concerning the long-term effects of prenatal stress, on fetal development [12]. MicroRNAs (miRNAs) are genome-encoded small non-coding RNAs of approximately 22 nucleotides in length and play an essential role in the post-transcriptional regulation of target gene expression [13–16]. miRNAs control gene expression post-transcriptionally by regulating mRNA translation or stability in the cytoplasm [17, 18]. Functional studies indicate that miRNAs are involved in critical biological processes during development and in cell physiology [13, 16]. Changes in their expression have been observed in several pathologies [16, 19].

Thus, miRNAs characterization has helped understand gene regulation and cellular proliferation, differentiation, and apoptosis and explain pathophysiology disorders, including kidney disorders [20–22]. Studies have reported that during kidney ontogeny miRNAs are indispensable for nephron development [23–26]. Moreover, underexpression of some miRNAs in MM progenitor cells reduces cell proliferation, resulting in early differentiation, and consequently, decreased number of nephrons [27, 28]. This phenomenon is characterized by increased apoptosis and high Bim expression in progenitor cells [27]. Thus, miRNAs modulate the balance between apoptosis and proliferation these metanephric primary cells [29].

We hypothesized that unknown epigenetic changes and miRNA expression profiling are associated with kidney developmental disorder in maternal protein-restricted offspring. Thus, we aimed to evaluate patterns of miRNA and predicted gene expression in the fetal kidney at 17 days of gestational (17-DG) protein-restricted male offspring to identify molecular

pathways and disorders involved in renal cell proliferation and differentiation during kidney development.

## Material and methodology

### Animal and diets

The experiments were conducted as described in detail previously [5, 6] on age-matched female and male rats of sibling-mated Wistar *HanUnib* rats (250–300 g) originated from a breeding stock supplied by CEMIB/ UNICAMP, Campinas, SP, Brazil. The environment and housing presented the right conditions for managing their health and well-being during the experimental procedure. Immediately after weaning at three weeks of age, animals were maintained under controlled temperature (25°C) and lighting conditions (07:00–19:00h) with free access to tap water and standard laboratory rodent chow (Purina Nuvital, Curitiba, PR, Brazil: Na+ content: 135 ± 3μEq/g; K+ content: 293 ± 5μEq/g), for 12 weeks before breeding. The Institutional Ethics Committee on Animal Use at São Paulo State University (#446-CEUA/UNESP) approved the experimental protocol, and the general guidelines established by the Brazilian College of Animal Experimentation were followed throughout the investigation. It was designated day 1 of pregnancy as the day in which the vaginal smear exhibited sperm. Then, dams were maintained ad libitum throughout the entire pregnancy on an isocaloric rodent laboratory chow with either standard protein content [NP, n = 36] (17% protein) or low protein content [LP, n = 51] (6% protein). The NP and LP maternal food consumption were determined daily (subsequently normalized for body weight), and the bodyweight of dams was recorded weekly in both groups. On 17 days of gestation (17-DG), the dams were anesthetized by ketamine (75mg/kg) and xylazine (10mg/kg), and the uterus was exposed. The fetuses were removed and immediately euthanized by decapitation. The fetuses were weighed and, the tail and limbs were collected for sexing. The metanephros was collected for Next Generation Sequencing (NGS), RT-qPCR, and immunohistochemistry analyses.

### Sexing determination

The present study was performed only in male 17-DG offspring, and the sexing was determined by Sry conventional PCR (Polymerase Chain Reaction) sequence analysis. The DNA was extracted by enzymatic lysis with proteinase K and Phenol-Chloroform. For reaction, the Master Mix Colorless—Promega was used, with the manufacturer's cycling conditions. The Integrated DNA Technologies (IDT) synthesized the primer following sequences bellow:

Forward: 5′-TACAGCCTGAGGACATATTA-3′

Reverse: 5′-GCACTTTAACCCTTCGATTAG-3′.

### Total RNA extraction

RNA was extracted from NP (n = 4) and LP (n = 4) whole kidneys using Trizol reagent (Invitrogen), according to the instructions specified by the manufacturer. Total RNA quantity was determined by the absorbance at 260 nm using a nanoVue spectrophotometer (GE Healthcare, USA). RNA Integrity was ensured by obtaining an RNA Integrity Number—RIN > 8 with Agilent 2100 Bioanalyzer (Agilent Technologies, Germany) [30].

### miRNA-Seq and data analysis

Sequencing was performed on the MiSeq platform (Illumina). The protocol followed the manufacturer's instructions available in (http://www.illumina.com/documents//products/datasheets/datasheet_truseq_sample_prep_kits.pdf). Briefly, the sequencing includes library

construction, and this was used 1μg total RNA. In this step, the adapters are connected, the 3 'and 5'. After ligation of adapters, a reverse transcription reaction was performed to create cDNA. It was then amplified by a standard PCR reaction, which uses primers containing a sequence index for sample identification—this cDNA library, subjected to agarose gel electrophoresis for miRNA isolation. After quantitation, the library concentration was normalized to 2 nM using 10 nM Tris-HCl, pH 8.5, and transcriptome sequencing was performed by MiSeq Reagent Kit v2 (50 cycles).

Data analysis was performed in collaboration with Tao Chen, Ph.D. from the Division of Genetic and Molecular Toxicological, National Center for Toxicological Research, Jefferson, AR, USA. The data from Next Generation Sequencing (NGS) of miRNAs were generated in FASTAQ format and imported into BaseSpace.com (Illumina, USA). The data quality was evaluated using the base calling CASAVA software developed by the manufacturer (Illumina). The analyzes were done by BaseSpace miRNA Analysis (from the University of Torino, Canada) and the sequence mapping of different miRNAs by Small RNA (Illumina, USA) for rat genome. The differentially expressed miRNA study was analyzed using Ingenuity Pathway Analysis software (Ingenuity, USA).

## miRNA expression validation

Four male offspring from different litters were used in each group for the miRNA (miR-127-3p, -144-3p, -298-5p, let7a-5p, -181a-5p, -181c-3p, and -199a-5p) expression analysis. Briefly, 450 ng RNA was reverse transcribed, without pre-amplification, using the TaqMan® MicroRNA Reverse Transcription Kit, according to the manufacturer's guidelines. Complementary DNA (cDNA) was amplified using TaqMan MicroRNA Assays (Life Technologies, USA) with *TaqMan® Universal PCR Master Mix*, *No AmpErase® UNG* (2x) on StepOnePlusTM Real-Time PCR System (Applied BiosystemsTM), according to the manufacturer's instructions. Data analysis was performed using relative gene expression evaluated using the comparative quantification method (Pfaffl, 2001). Based on stability analysis, the U6 snRNA and U87 scaRNA was used as a reference gene. All relative quantifications were evaluated using the DataAssist software, v 3.0, using the ΔΔCT method. miRNA data have been generated following the MIQE guidelines [31].

## RT-qPCR of predicted target genes

For the cDNA synthesis, the High Capacity cDNA reverse transcription kit (Life Technologies, USA) was used. The RT-qPCR reactions for Bax, Bim, Caspase-3, Collagen 1, GDNF, PCNA, TGFβ-1, Bcl-2, Bcl-6, c-Myc, c-ret, cyclin A, Map2k2, PRDM1, Six-2, Ki-67, MTOR, β-catenin, ZEB1, ZEB2, NOTCH1, and IGF1 gene was performed by SYBR Green Master Mix (Life Technologies, USA) provided by IDT® Integrated DNA Technologies (Table 1). The reactions were done in a total volume of 20 μL using 2 μL of cDNA (diluted 1:30), 10μL SYBER Green Master Mix (Life Technologies, USA), and 4 μL of each specific primer (5 nM). Amplification and detection were performed using the StepOnePlusTM Real-Time PCR System (Applied BiosystemsTM). Ct values were converted to relative expression values using the ΔΔCt method with offspring metanephros data normalized with GAPDH as a reference gene [32].

## Immunohistochemistry

The fetus (n = 4 per group) was removed and immediately fixed in 4% paraformaldehyde (0.1 M phosphate, pH 7.4). The materials were dehydrated, diaphanized, and included in paraplast, and the blocks were cut into 5-μm-thickness sections. Histological sections were deparaffinized and processed for immunofluorescence and immunoperoxidase. The sections were

**Table 1. Dilution of antibodies used in immunohistochemistry.**

| GENE | FORWARD SEQUENCE | REVERSE SEQUENCE |
|---|---|---|
| Six-2 | 5'-GCCGAGGCCAAGGAAAGGGAG-3' | 5'-GAGTGGTCTGGCGTCCCCGA-3' |
| c-myc | 5'-AGCGTCCGAGTGCATCGACC-3' | 5'-ACGTTCCAAGACGTTGTGTG-3' |
| c-ret | 5'-GTTTCCCTGATGAGAAGAAGTG-3' | 5'-GTGGACAGCAGGACAGATA-3' |
| Bcl-2 | 5'-ACGGTGGTGGAGGAACTCTT-3' | 5'-GTCATCCACAGAGCGATGTTG-3' |
| Col-1 | 5`-ACCTGTGTGTTCCCCACT-3` | 5`-CTTCTCCTTGGGGTTTGGGC-3` |
| TGFβ-1 | 5`-GGACTCTCCACCTGCAAGAC-3` | 5`-GACTGGCGAGCCTTAGTTTG-3` |
| Ciclin A | 5'-GCC TTCACCATTCATGTGGAT-3' | 5'-TTGCTGCGGGTAAAGAGACAG-3' |
| Bax | 5'-TTCAGTGAGACAGGAGCTGG-3' | 5'-GCATCTTCCTTGCCTGTGAT-3' |
| Bim | 5'-CAATGAGACTTACACGAGGAGG-3' | 5'CCAGACCAGACGGAAGATGAA-3' |
| Casp 3 | 5'-ACGGGACTTGGAAAGCATC-3' | 5'-TAAGGAAGCCTGGAGCACAG-3' |
| GDNF | 5'-CAGAGGGAAAGGTCGCAGAG-3' | 5'-TCGTAGCCCAAACCCAAGTC-3' |
| Ki-67 | 5'- GTCTCTTGGCACTCACAG-3' | 5'-TGGTGGAGTTACTCCAGGAGAC-3' |
| mTOR | 5`-ACGCCTGCCATACTTGAGTC-3` | 5`-TGGATCTCCAGCTCTCCGAA-3` |
| VEGF | 5`-CGGGCCTCTGAAACCATGAA-3` | 5`-GCTTTCTGCTCCCCTTCTGT-3` |
| GAPDH | 5`-CAACTCCCTCAAGATTGTCAGCAA-3` | 5`-GGCATGGACTGTGGTCATGA-3` |
| B-catenin | 5'-AGTCCTTTATGAGTGGGAGCAA-3' | 5'- GTTTCAGCATCTGTGACGGTTC-3' |
| Map2K2 | 5'- ACCGGCACTCACTATCAACC-3' | 5'-TTGAGCTCACCGACCTTAGC-3' |
| Bcl-6 | 5'-CCAACCTGAAGACCCACACTC-3' | 5'-GCGCAGATGGCTCTTCAGAGTC-3' |
| PCNA | 5'-TTTGAGGCACGCCTGATCC-3' | 5'-GGAGACGTGAGACGAGTCCAT-3' |
| PRDM1 | 5'-CTTGTGTGGTATTGTCGGGAC-3' | 5'-CACGCTGTACTCTCTCTTGG-3' |
| NOTCH1 | 5`-ACTGCCCTCTGCCCTATACA-3` | 5`-GACACGGGCTTTTCACACAC-3` |
| IGF1 | 5`-AAGCCTACAAAGTCAGCTCG-3` | 5`-GGTCTTGTTTCCTGCACTTC-3` |
| ZEB1 | 5'-CATTTGATTGAGCACATGCG-3' | 5'-AGCGGTGATTCATGTGTTGAG-3' |
| ZEB2 | 5'-CCCTTCTGCGACATAAATACGA-3' | 5'-TGTGATTCATGTGCTGCGAGT-3' |

incubated with a blocking solution (8% fetal bovine serum, 2.5% bovine albumin, and 2% skimmed milk powder in PBS). Subsequently, set with the primary antibody (anti-Six-2) diluted in PBS containing 1% skim milk overnight under refrigeration. After washing with PBS, the sections were incubated with a specific secondary antibody, conjugated to the Alexa 488 fluorophore, diluted in the same buffer, containing 1% milk for 2 hours at room temperature. After successive washes with PBS, the slides were mounted with coverslips using the Vectashield fluorescent assembly medium (Vector Laboratories, Inc. Burlingame). The fluorescence in the specimen was detected by laser confocal microscopy. The images were obtained using the Focus Imagecorder Plus system. For the c-Myc, Ki-67, Bcl-2, TGFβ-1, β-catenin, ZEB1, ZEB2, Caspase 3 cleaved, cyclin A and WT1 proteins, immunohistochemistry was performed. The slides were hydrated, and after being washed in PBS pH 7.2 for 5 minutes, the antigenic recovery was made with citrate buffer pH 6.0 for 25 minutes in the pressure cooker. The slides were washed in PBS. Subsequently, endogenous peroxidase blockade with hydrogen peroxide and methanol was performed for 10 minutes in the dark. The sections were rewashed in PBS. Blocking of non-specific binding was then followed, and the slides were incubated with a blocking solution (5% skimmed milk powder, in PBS) for 1 hour. The sections were incubated with the primary antibody (Table 2), diluted in 1% BSA overnight in the refrigerator. After washing with PBS, the sections were exposed to the specific secondary antibody for 2 hours at room temperature. The slides were washed with PBS. The slices were revealed with DAB (3,3'- diaminobenzidine tetrahydrochloride, Sigma—Aldrich CO®, USA). After successive washing in running water, the slides were counterstained with hematoxylin, dehydrated, and mounted with a coverslip, using Entellan®. The images were obtained using

**Table 2. Sequence of the primers used for RT-qPCR, designed by the company IDT.**

| Antibody | Dilution | Company |
|---|---|---|
| Anti-Six-2 (11562-1-AP) | 1:50 | Proteintech |
| Anti-c-Myc (NBP1-19671) | 1:150 | Novus Biologicals |
| Anti-Ki-67 (ab16667) | 1:100 | Abcam |
| Anti-Bcl-2 (ab7973) | 1:100 | Abcam |
| Anti-TGFβ-1 (sc-146) | 1:50 | Santa Cruz |
| Anti-B-catenina (ab32572) | 1:500 | Abcam |
| Anti-ZEB1 (sc-10572) | 1:50 | Santa Cruz |
| Anti-ZEB2 (sc-48789) | 1:50 | Santa Cruz |
| Anti-VEGF (NB100-664) | 1:50 | Novus Biologicals |
| Anti-Caspase-3 clivada (9664) | 1:200 | Cell Signaling |
| Anti-Ciclina A (sc-31085) | 1:50 | Santa Cruz |
| Anti-WT1 (sc-192) | 1:50 | Santa Cruz |
| Anti-mTOR (cs-7c10) | 1:200 | Cell Signaling |

the photomicroscope (Olympus BX51) or a Zeiss LSM 780-NLO confocal on an Axio Observer Z.1 microscope (Carl Zeiss AG, Germany) from the National Institute of Science and Technology on Photonics Applied to Cell Biology (INFABIC) at the State University of Campinas.

## Morphology quantification

Paraffin 5 μm kidney sections were analyzed using CellSens Dimension software from a photomicroscope (Olympus BX51). The kidney slices were accessed to determine the nephrogenic area, CM and UB protein and cell number, hematoxylin-eosin stained in 17-DG LP fetus (n = 5) compared to age-matched NP offspring (n = 5) from different mothers. We quantified all CM and UB of each metanephros analyzed (4NP and 4LP from different mothers), and statistical analysis was performed by t-test, and the values were expressed as mean ± SD. The p≤0.05 was considered significant. GraphPad Prism v01 Software, Inc., USA, was used for statistical analysis and figure construction.

## Statistical analysis

The t-test was used, and the values were expressed as mean ± standard deviation (SD). P≤0.05 was considered significant. GraphPad Prisma v. 01 software (GraphPad Software, Inc., USA) was used for statistical analysis and figure construction.

## Results

### Expression of miRNAs by miRNA-Seq

To understand the microRNA changes associated with maternal low-protein renal programming, we performed the expression of a global miRNA profiling analysis. It was identified 44 deregulated miRNAs (p ≤ 0.05), of which 19 and 25 miRNAs, respectively, were up-or down-regulated (Table 3). The top expressed miRNAs and their functions, pathways, and networks were identified using Ingenuity Software (Table 4).

### Validation of miRNA expression

*In* the LP group's animals, Let-7a-5p, miR-181a-5p, miR-181c-3p were upregulated, while the miR-127-3p, miR-144-3p, and miR-199a-5p were downregulated relative to NP animals. The

**Table 3. Lists of the deregulated miRNAs obtained by miRNA-Seq.**

| miRNAs up-regulated | FC | miRNAs down-regulated | FC |
|---|---|---|---|
| 83_ACCACCAACCGTTGACTGTACC_rno-mir-181a-2 | 1.55 | 69_ACAGTAGTCTGCACATTGGTT_rno-mir-199a | 0.76 |
| 38_AACATTCAACGCTGTCGGTG_rno-mir-181a-2 | 2.08 | rno-miR-136-3p | 0.53 |
| 10_GGCAGAGGAGGGCTGTTCTTCC_rno-mir-298 | 1.44 | rno-let-7g-5p | 0.78 |
| rno-miR-298-5p | 1.47 | rno-miR-144-3p | 0.49 |
| rno-miR-183-5p | 1.46 | 13_AAGGGATTCTGATGTTGGTCACACTC_rno-mir-541 | 0.52 |
| rno-miR-181d-5p | 1.36 | 15_TCCCTGAGGAGCCCTTTGAGCCTGAAA_rno-mir-351-2 | 0.50 |
| rno-miR-455-3p | 1.93 | 56_TCGGATCCGTCTGAGCTTGGC_rno-mir-127 | 0.53 |
| 35_AACATTCATTGCTGTCGGTGGGA_rno-mir-181b-1 | 1.89 | 9_ATATAATACAACCTGCTAAGTGT_rno-mir-374 | 0.54 |
| 50_CAGTGCAATGATGAAAGGGC_rno-mir-130b | 1.85 | 56_TCGGATCCGTCTGAGCTTGG_rno-mir-127 | 0.52 |
| 83_ACCACCAACCGTTGACTGTACCT_rno-mir-181a-2 | 1.81 | rno-miR-320-3p | 0.55 |
| rno-miR-151-3p | 1.31 | rno-miR-376b-3p | 0.56 |
| rno-miR-181c-3p | 1.36 | 16_AAACCGTTACCATTACTGAGTTT_rno-mir-451 | 0.56 |
| 69_TACAGCAGGCACAGACAGGCAGT_rno-mir-214 | 1.53 | 56_TCGGTCGATCGGTCGGTCGGTT_rno-mir-341 | 0.56 |
| rno-miR-195-3p | 1.58 | 16_AAACCGTTACCATTACTGAGTTTAGT_rno-mir-451 | 0.54 |
| 21_TACCCTGTGTAGATCCGAATTTGTGA_rno-mir-10a | 1.2 | 12_GGATATCATCATATACTGTAAG_rno-mir-144 | 0.59 |
| rno-miR-1298 | 1.52 | 60_TCAGTGCATCACAGAACTTTGTTT_rno-mir-148b | 0.72 |
| rno-miR-92b-3p | 1.36 | 15_TCCCTGAGGAGCCCTTTGAGCCTGT_rno-mir-351-2 | 0.77 |
| 61_ACCACAGGGTAGAACCACGGAA_rno-mir-140 | 1.60 | 13_AAGGGATTCTGATGTTGGTCACAC_rno-mir-541 | 0.58 |
| 50_CAGTGCAATGATGAAAGGGCATA_rno-mir-130b | 1.31 | 15_CTGAGAACTGAATTCCATGGGTT_rno-mir-146a | 0.58 |
| | | 9_GACCCTGGTCTGCACTCTGTCT_rno-mir-504 | 0.58 |
| | | rno-let-7b-5p | 0.69 |
| | | rno-let-7f-5p | 081 |
| | | 14_TCCCTGAGACCCTTTAACCTG_rno-mir-125a | 0.58 |
| | | rno-miR-410-3p | 0.63 |
| | | rno-miR-541-5p | 0.68 |

**Table 4. Top canonical pathways affected by differentially expressed miRNAs in 17 DG LP metanephros.**

| 17 DG | Pathway analysis results (IPA) | Number of miRNAs | p-value/score |
|---|---|---|---|
| NP vs LP | **Top Molecular and Cellular Functions** | | |
| | Cellular Development | 12 | 4.71E-02–4.31E-05 |
| | Cellular Growth and Proliferation | 12 | 3.41E-02–1.60E-04 |
| | DNA Replication, Recombination, and Repair | 04 | 3.12E-02–4.83E-04 |
| | Cell Cycle | 04 | 3.49E-02–7.49E-04 |
| | Cellular Movement | 06 | 3.49E-02–1.11E-03 |
| | **Top Networks** | | |
| | Organismal Injury and Abnormalities, Reproductive System Disease, Cancer | | 34 |
| | **Top Tox Lists** | | |
| | Renal Ischemia-Reperfusion Injury microRNA Biomarker Panel (Mouse)<br>**Top 10 highly expressed miRNAs**<br>miR-199a-5p; miR-136-3p<br>miR-181a-5p; miR-298-5p<br>miR-144-3p; miR-541-5p<br>miR-127-3p; miR-374b-5p<br>miR-183-5p; Let-7a-5p | | 4.31E-05 |

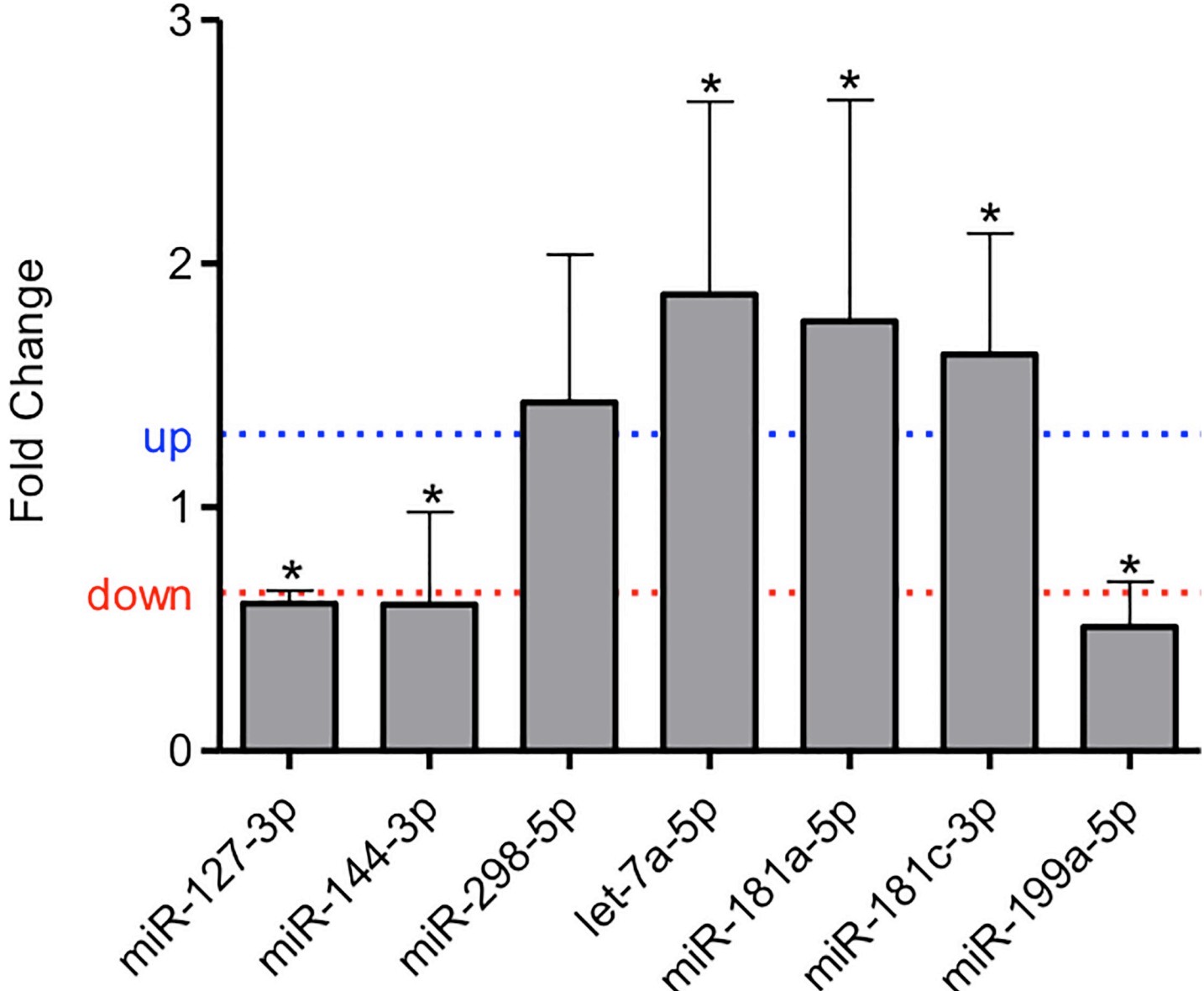

**Fig 1. Expression of miRNAs in the metanephros from the 17th day LP fetus compared to their expression level in the control group.** Reference genes U6 and U87, protein complexes composed of small nuclear RNAs (snRNAs), were used to normalize each miRNA expression. The authors established a cutoff point variation of 1.3 (upwards) or 0.65 (downwards) and data are expressed as fold change (mean ± SD, n = 4) concerning the control group. * p≤0.05: statistical significance versus NP.

results do not show any difference in miR-298 expression, comparing both groups (Fig 1). Table 5 revealed the values obtained by miRNAs sequencing with the RT-qPCR validation data. Although significant miRNA expression difference was observed in LP relative to NP offspring, the fold change (FC) of the validated miRNAs was similar to both techniques.

## miRNA-gene targets

The expression genes of predicted targets of different miRNA such as Six-2, Bcl-2, PRDM1, cyclin A, PCNA, GDNF, Collagen 1, Caspase 3, and Bim in LP did not differ significantly from NP fetus. However, Bax, TGFβ-1 Bcl-6, c-ret, Map2k2, Ki-67, mTOR, β-catenin, ZEB1, ZEB2,

**Table 5. Comparison between the values obtained in the miRNA sequencing and the validation by RT-qPCR.**

| miRNA (17 DG) | log FC Sequencing | Fold Change (FC) | *p-value* | miRNA (17 DG) | Fold Change (FC) | log FC qPCR | *p-value* |
|---|---|---|---|---|---|---|---|
| miR-127-3p | -0.911189177 | 0.5317 | 0.01053498 | miR-127-3p | 0.6045 | -0.7262 | 1.97E-08 |
| miR-144-3p | -1.024909088 | 0.4914 | 0.00754482 | miR-144-3p | 0.6014 | -0.7335 | 0.0321508 |
| miR-298-5p | 0.555324736 | 1.4695 | 0.0083628 | miR-298-5p | 1.4317 | 0.5177 | 0.0648687 |
| Let-7a-5p | | | | Let-7a-5p | 1.8747 | 0.9067 | 0.0106146 |
| miR-181a-5p | 0.637181521 | 1.5553 | 0.00520696 | miR-181a-5p | 1.7645 | 0.8193 | 0.0354613 |
| miR-181c-3p | 0.40731742 | 1.3262 | 0.02187935 | miR-181c-3p | 1.6265 | 0.7018 | 0.0273168 |
| miR-199a-5p | -0.401388293 | 0.7571 | 0.00193047 | miR-199a-5p | 0.5086 | -0.9755 | 4.551E-05 |

and IGF1 gene expression were upregulated in the 17-DG LP group compared to age-matched controls. Conversely, c-Myc, and NOTHC1 were downregulated in maternal protein-restricted offspring (Fig 2).

## Fetus body mass and metanephros morphometry

The 17-DG LP body mass did not differ from the age-matched NP offspring. However, LP's metanephros mesenchyme showed a 7.6% reduced area and a 29% reduction in the cortex thickness than the NP group (Fig 3).

## Immunohistochemistry

In the present study, the LP fetus showed a significant reduction (about 69%) of Six-2 cap fluorescence than NP offspring (Fig 4).

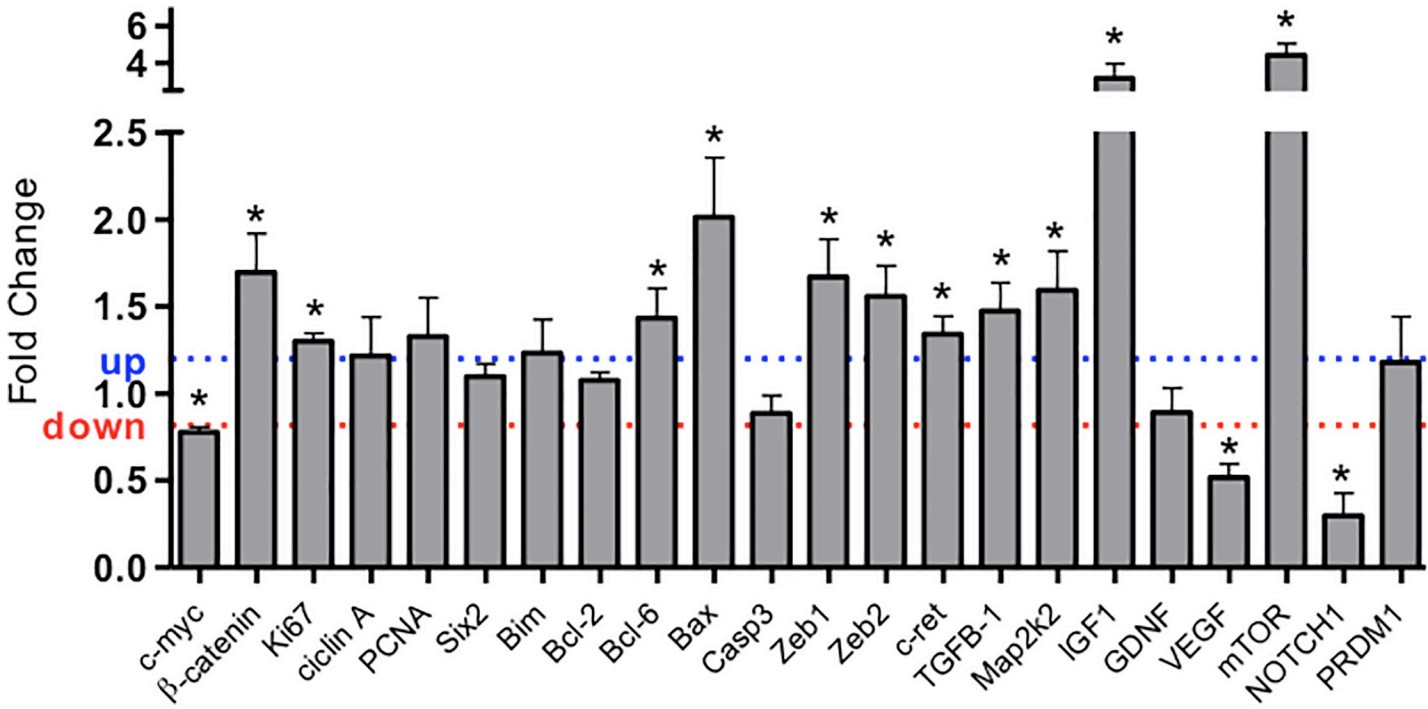

**Fig 2. Expression of mRNA estimated by SyBR green RT-qPCR of metanephros from the 17th day LP fetus.** The expression was normalized with GAPDH. The authors established a cutoff point variation of 1.3 (upwards) or 0.65 (downwards) and data are expressed as fold change (mean ± SD, n = 4) concerning the control group. * p≤0.05: statistical significance versus NP.

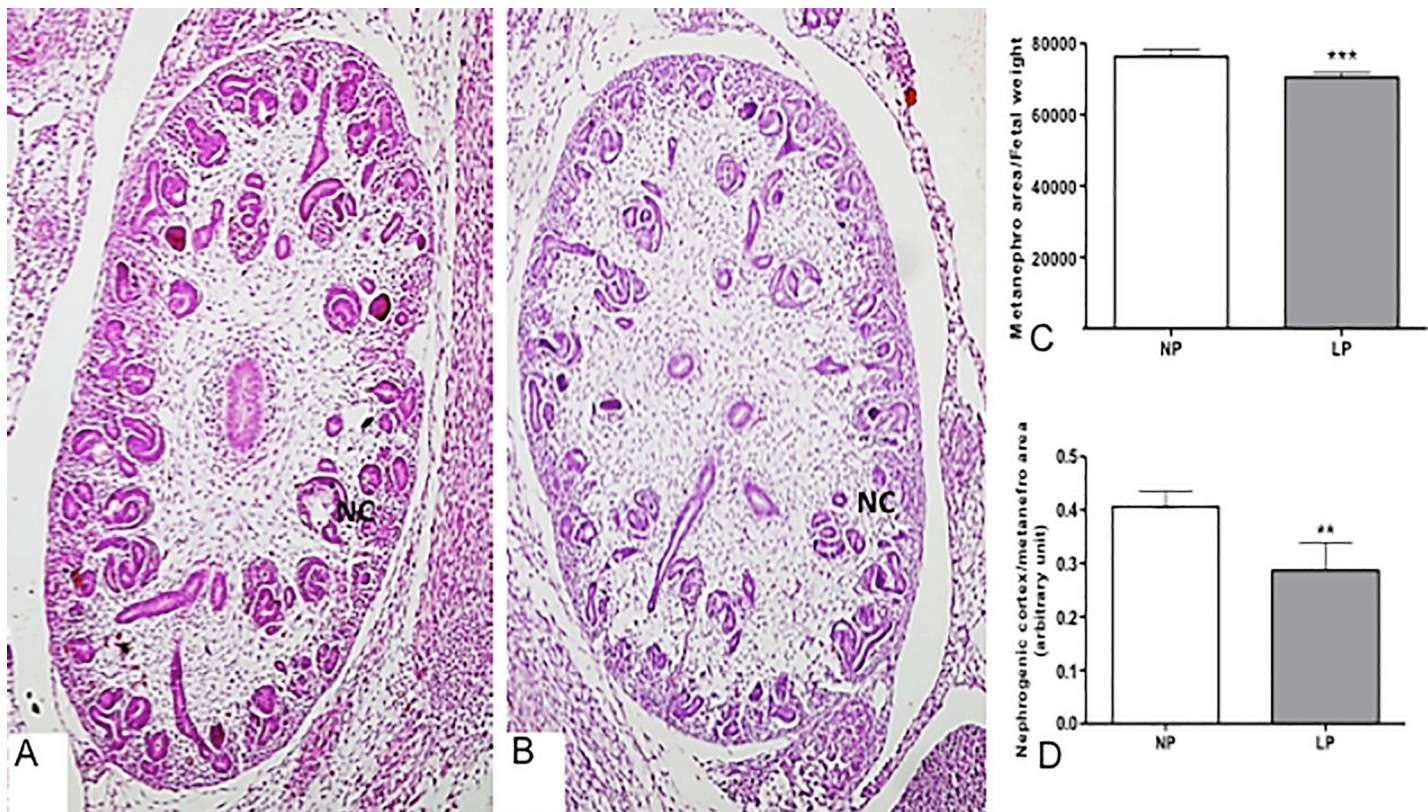

**Fig 3. Metanephros of the fetus with 17 DG and quantifications.** Comparing the HE stained micrography, we can observe the difference in the NP (A) and LP(B) metanephros size and nephrogenic cortex (NC) thickness. The differences between these parameters are statistically significant (C and D). **p<0.005; ***p<0.0001.

The Six-2 immunoperoxidase analysis demonstrated a reduced cell number (14%) in LP CM from associated with 28% reduced Six-2+ cells relative to the cap area compared to NP offspring (Fig 4). The present study also showed a significant percent reduction of c-Myc CM and UB immunostained cells (less 14%) in LP relative to NP offspring (Fig 4). Additionally, the percentage of Ki-67 labeled area in CM was 48% lesser in LP compared to NP fetus, while Bcl-2 and cleaved caspase-3 immunoreactivity were not different from both groups (Figs 5 and 6).

The present study also showed a significant percent reduction of c-Myc CM and UB immunostained cells (less 14%) in LP relative to NP offspring (Fig 4). Additionally, the percentage of Ki-67 labeled area in CM was 48% lesser in LP compared to NP fetus, while Bcl-2 and cleaved caspase-3 immunoreactivity were not different from both groups (Figs 5 and 6). On the other hand, in LP, the CM and UB β-catenin labeled area were 154 and 85% raised, respectively, compared to that available in NP offspring (Fig 7).

At the same time, mTOR immunoreactivity distribution also occupied a significantly more extensive area in LP CM (139%) and UB (104%) than in the NP fetus (Fig 7). In the LP offspring, the TGFβ-1 in UBs cells staining increased (about 30%), while in the CM, the immunostained cells were not different related to the NP group (Fig 8).

The ZEB1 metanephros-stained, located in the CM nuclei cells, enhanced 30% in LP compared to the NP fetus (Fig 8). Simultaneously, the ZEB2 immunofluorescence, although present in whole metanephros structures, was similar in both experimental groups (Fig 8). The current study, taking into account miRNA and mRNA expression and proteins

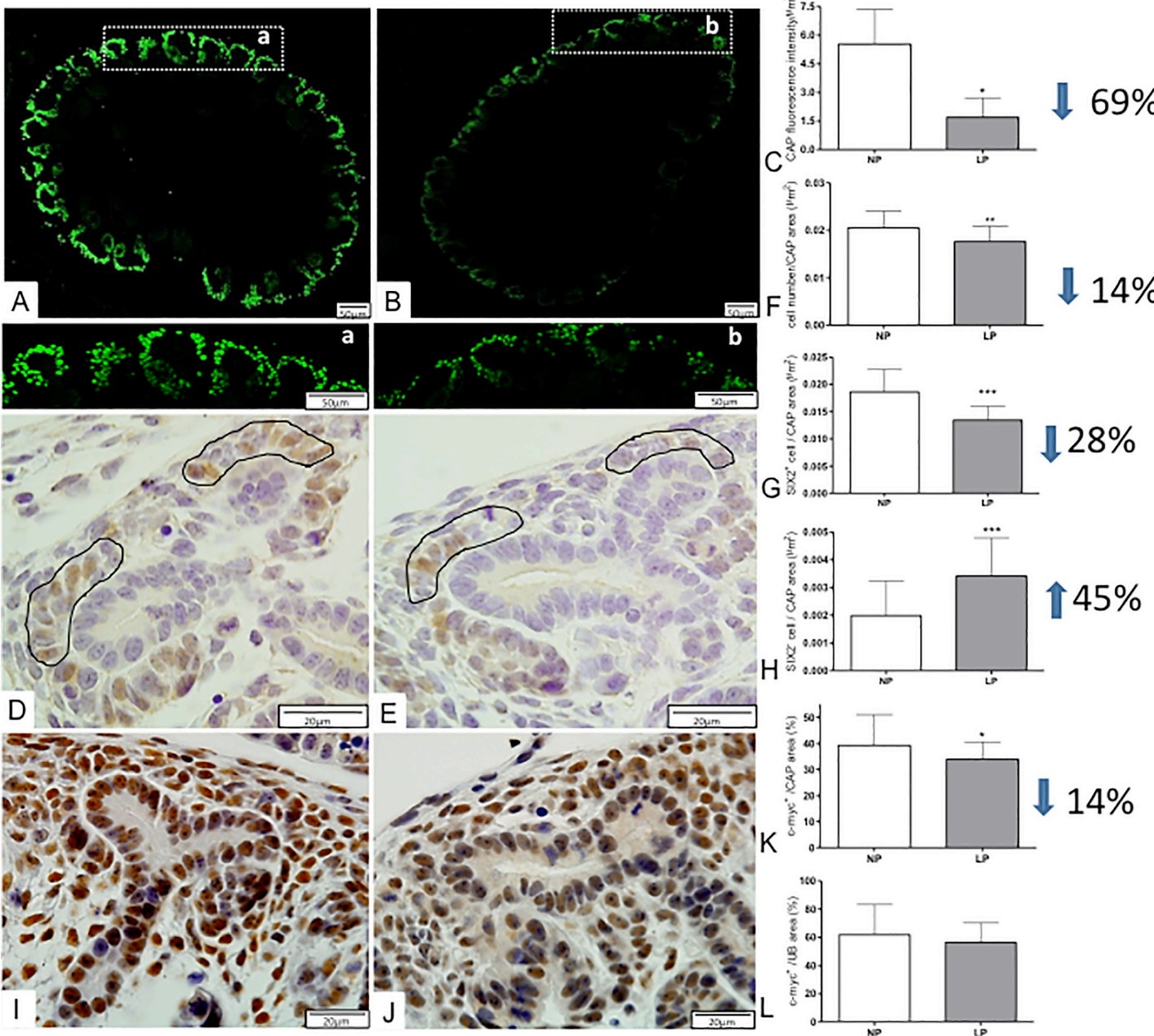

**Fig 4. Immunofluorescence and immunoperoxidase for Six-2 and c-Myc in metanephros of 17DG fetus.** The Six-2 immunomarker cells in LP (A, a) was significantly reduced when compared to NP (B,b, C) in metanephros. Additionally, the Six-2 immunostained cells were significantly reduced in LP (E) caps (circled by black lines; F, G) when compared to NP (D). On the other hand, the c-Myc labeled area was reduced in the LP cap (K) but was the same in UB (L) when compared to NP (J). *p<0.005;**p<0.001; ***p<0.0001.

immunostaining results present above may permit schedule representative pathway interactions to explain the experimental findings (Fig 9).

## Discussion

Knowledge about cellular and molecular mechanisms of nephrogenesis has increased [33–37]. However, many regulatory factors and signaling pathways involved in renal ontogenesis

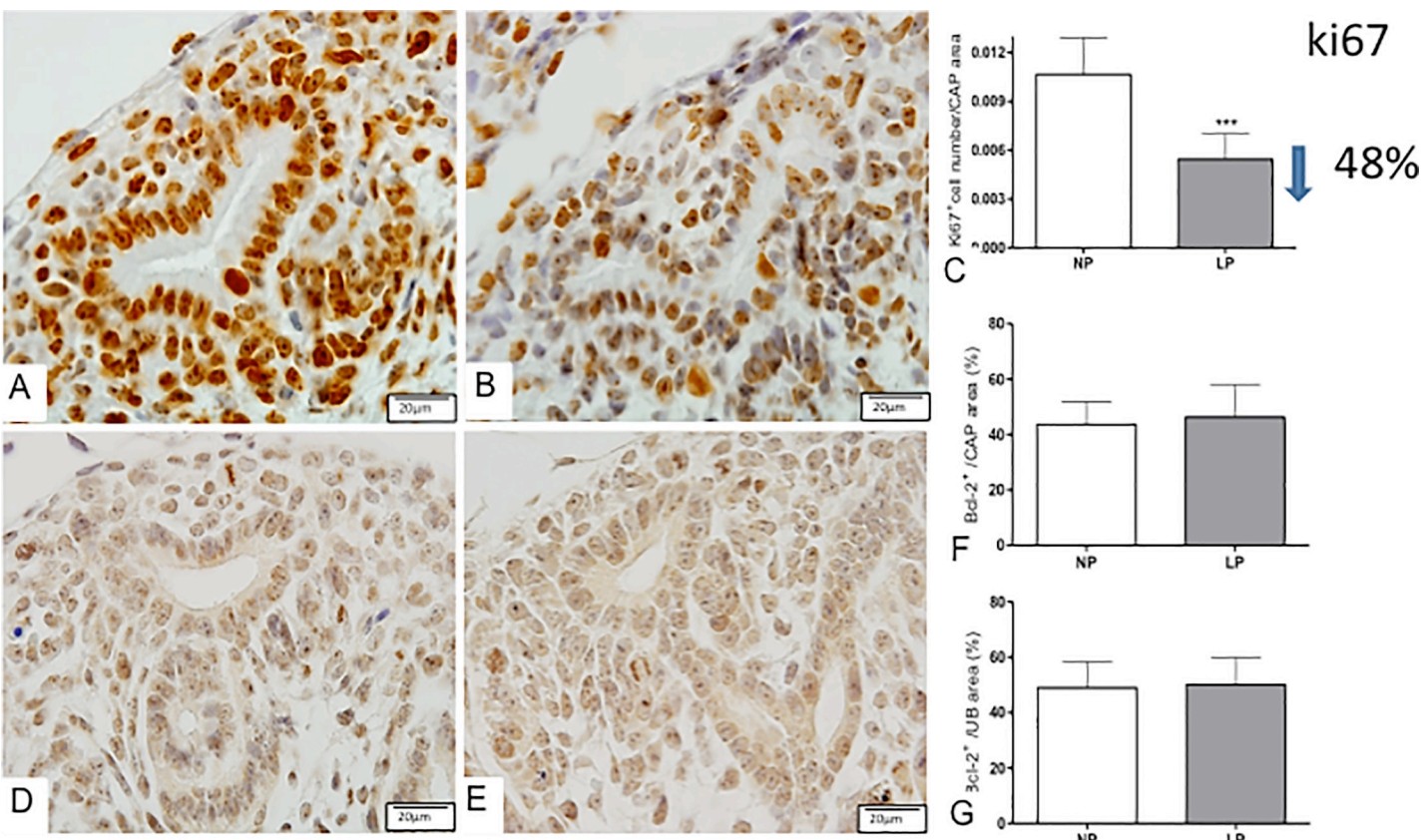

**Fig 5. Immunoperoxidase for Ki-67 and Bcl-2 in metanephros of 17DG fetus.** The number of Ki-67+ cells was reduced in LP (B) when compared to NP (A) and was statistically significant in the caps (C). The Bcl-2 labeled area was the same in NP (D) and LP (E) in both UB and cap (F, G). ***p<0.0001.

remain unclear [38]. miRNAs play a crucial role in regulating gene expression during renal development [25, 39–41]. To our knowledge, miRNA and mRNA expression analyses in maternal LP intake 17-DG male mesenchyme cells have not been performed.

We propose a novel molecular mechanism involved in inhibiting early nephrogenesis, resulting in a reduced number of nephrons. We used NGS to evaluate miRNA expression and found that 19 miRNAs were up- and 25 downregulated in 17-DG LP compared to NP meta-nephros. Among the top 10 deregulated miRNAs, we selected 7 miRNAs with biological targets involved in proliferation, differentiation, and cellular apoptosis. Both miRNA-Seq and TaqMan data analysis revealed consistent and specific changes in miRNA expression in LP animals relative to control NP age-matched animals.

The miR-181 family is composed of four highly conserved members, namely miR-181a, miR-181b, miR-181c, and miR-181d [42]. In neoplastic cells, miR-181a acts as a tumor suppressor, inhibiting cellular proliferation and migration and inducing cellular apoptosis [43]. This study revealed increased expression of miR-181a-5p in 17-DG LP relative to age-matched NP offspring. Although caspase mRNA expression was unaltered, a two-fold increase in Bax/Bcl-2 mRNA ratio in LP compared to NP offspring suggests increased apoptosis in the CM, indicating that apoptosis is regulated post-transcriptionally. Studies have shown that the BCL family promotes cytochrome release from the mitochondria and then inhibits the activation of Casp3, thereby inhibiting cellular apoptosis [44]. Li et al. used an acute lung injury model to reveal that overexpressed miR-181a is related to decreased Bcl-2 protein level; conversely, miR-181a inhibition increased Bcl-2 levels [45]. This study confirmed the results of Lv et al.,

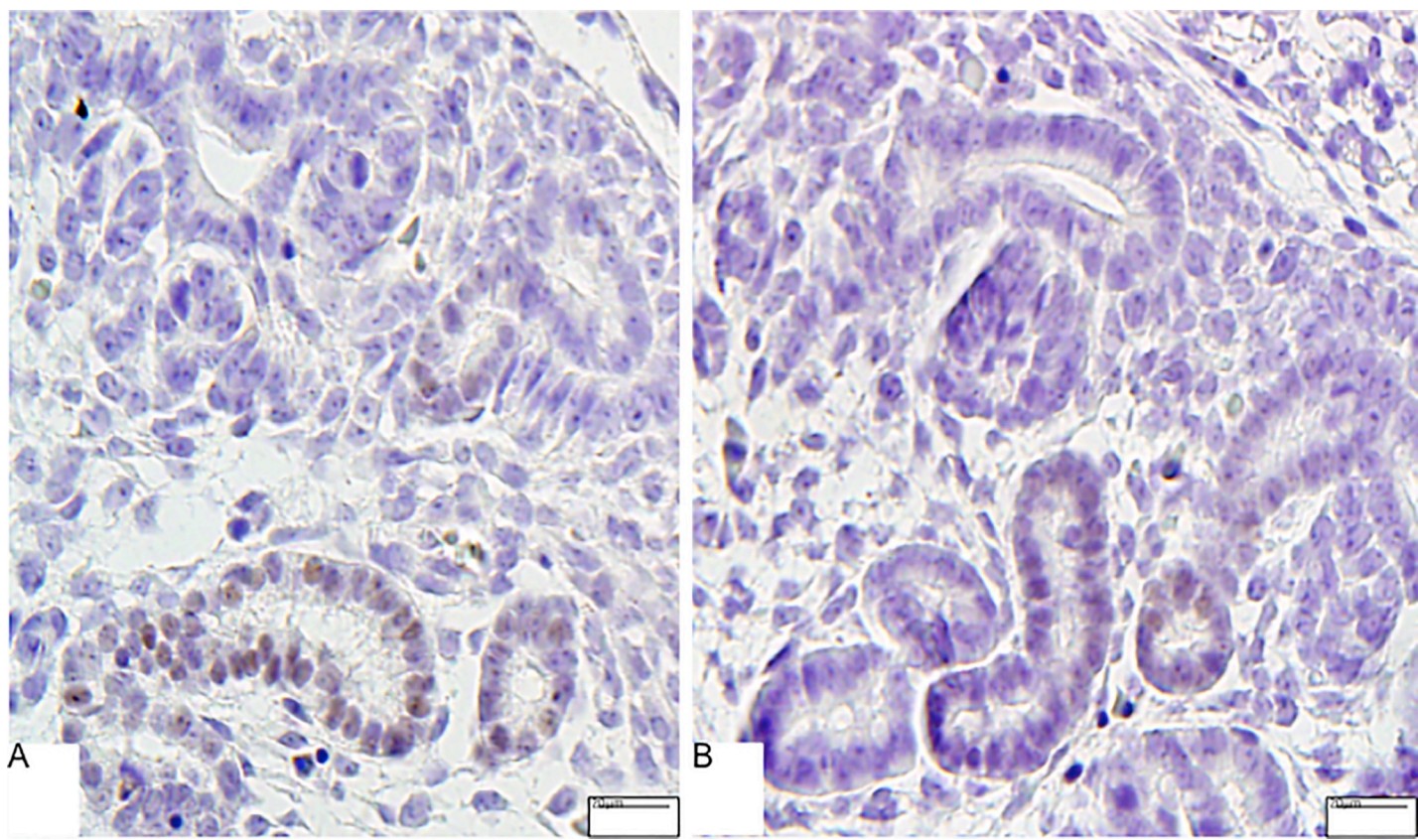

**Fig 6. Immunoperoxidase for cleaved caspase 3 in metanephros of 17DG fetus.** The immunostained cells were preferentially located in the ureteric epithelium. However, the quantity is not different in LP (B) than in NP (A).

who demonstrated that miR-181c regulates the expression of Six-2 expression and cell proliferation negatively, parallel to the loss of mesenchymal cells phenotype during kidney development in LP 17-DG offspring [25].

Xiang et al. demonstrated that increased miR-144 expression suppresses renal carcinoma proliferation, resulting in a shorter G2/M phase. Moreover, Xiang et al. revealed that overexpression of miR-144 inhibits mTOR gene and protein expression [46]. Nijland et al. demonstrated that an increase in mTOR signaling is crucial for determining the number of nephrons in embryos whose mothers were subjected to nutrient restriction [47]. Mammalian target of rapamycin complex 1 (mTORC1) is essential for embryo development; however, how this complex regulates the balance between growth and autophagy under physiological conditions and environmental stress remains unknown [48]. Therefore, mTOR signaling may be involved in cellular responses in animals exposed to LP intake during gestation; in the perception, induction, and termination of autophagy; and in response to intracellular nutrient availability [46]. Hypothetically, during severe protein restriction, reduced expression of miR-144-3p may be associated with increased mTOR expression, approximately 139% and 104% in CM cells and UB, respectively, to compensate for the loss of nephrons in the 17-DG LP offspring.

Chen et al. defined miR-127 as a new regulator of cell senescence through Bcl-6 [49]. Pan et al. reported that miR-127 underexpression correlates with increased cell proliferation in liver cells [50]. This study showed a decrease in cell proliferation and a significant reduction of cells positively labeled for Ki-67 in the CM of protein-restricted animals. Moreover, a reduction in nephrogenic area and proliferation in LP progeny was observed, which was consistent

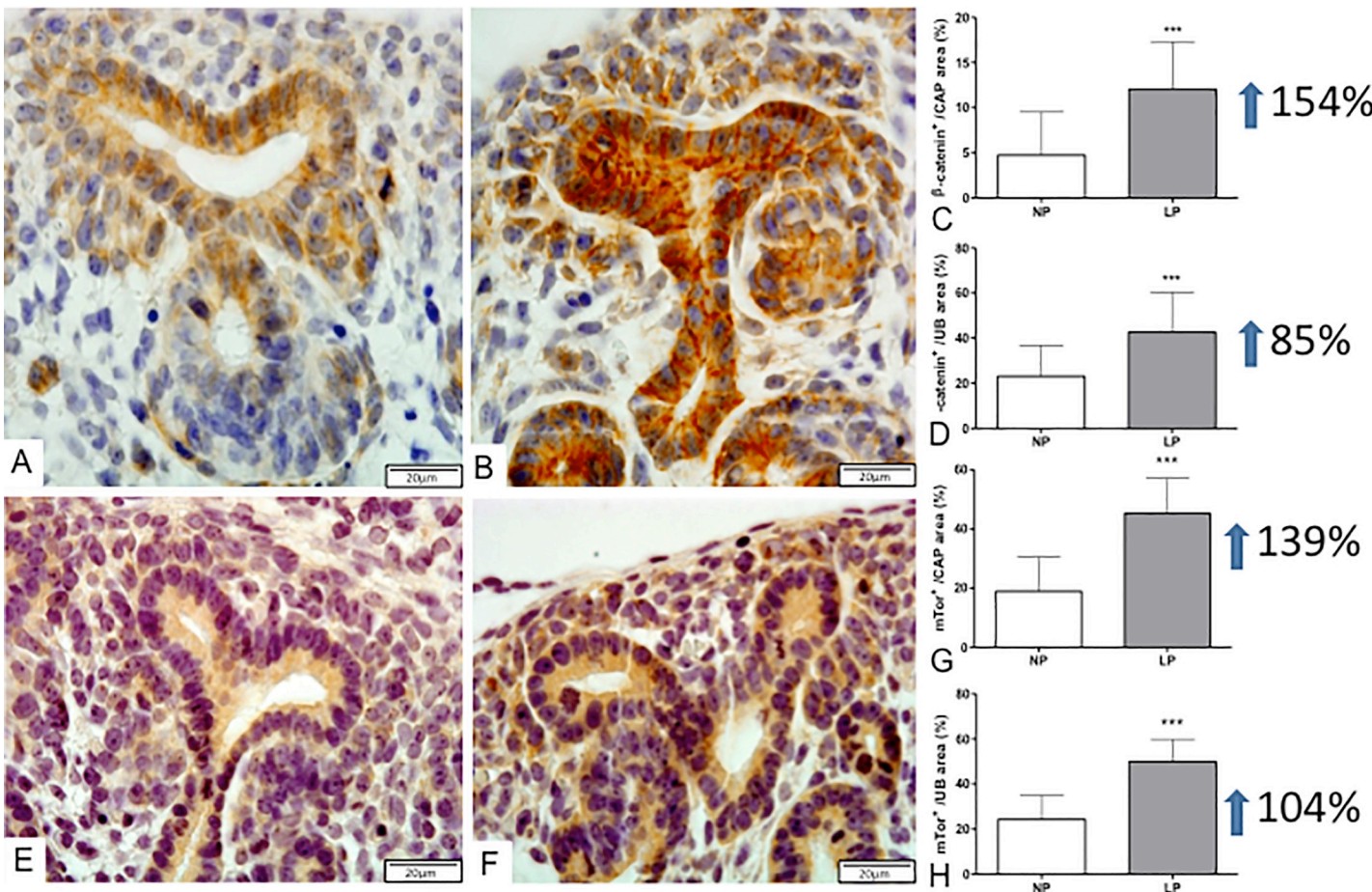

**Fig 7. Immunoperoxidase for b-catenin and mTor in metanephros of 17DG fetus.** In LP (B), the b-catenin labeled area was significantly raised in both CAP cells and UB epithelia (C, D) when compared to NP offspring (A). Also, mTor immunoreactivity occupied a more extensive area in LP (F) than in NP (E) offspring in analyzed structures (G, H). ***P<0.0001.

with the results of Menendez-Castro et al. in 8.4% protein-restricted progeny [51, 52]. Thus, increased Ki-67 and Bcl-6 mRNA expression, accompanied with reduced miR-127-3p expression in the 17-DG LP cap could be associated with counter-regulatory mechanisms to maintain proliferation.

Sun et al. demonstrated that overexpression of miR-199a-5p reduces cystic cell proliferation and induces apoptosis, in addition to controlling the cell cycle [53]. In this study, expression of miR-199a-5p is reduced in 17-DG LP is accompanied by the increased transcription of Ki-67, a cellular proliferation marker, and Map2k2 is associated with decreased Ki-67 reactivity in LP 17-DG metanephros. Thus, gestational undernutrition promotes differentiation through a post-transcriptional mechanism. Notably, our results reveal a repressive role of zinc-finger E-box binding homeobox 1 (ZEB1), an EMT inducer, which maintains stem cell pluripotency during embryonic stem cell differentiation. β-catenin is known to activate nuclear ZEB1 transcription resulting in ZEB1 expression [34]. The TGFβ signaling pathway, one of the best-studied pathways, can induce EMT during embryonic development. Several TGFβ like ligands are required for embryonic development. However, not all TGFβ-mediated effects on EMT depend on ZEB1/2, knockout cells can induce the expression of the mesenchymal genes fibronectin and N-cadherin. However, E-cadherin is no longer downregulated, and actin fibers are also formed [54]. Karner et al. reported that during renal development, the Wnt9b/β-catenin

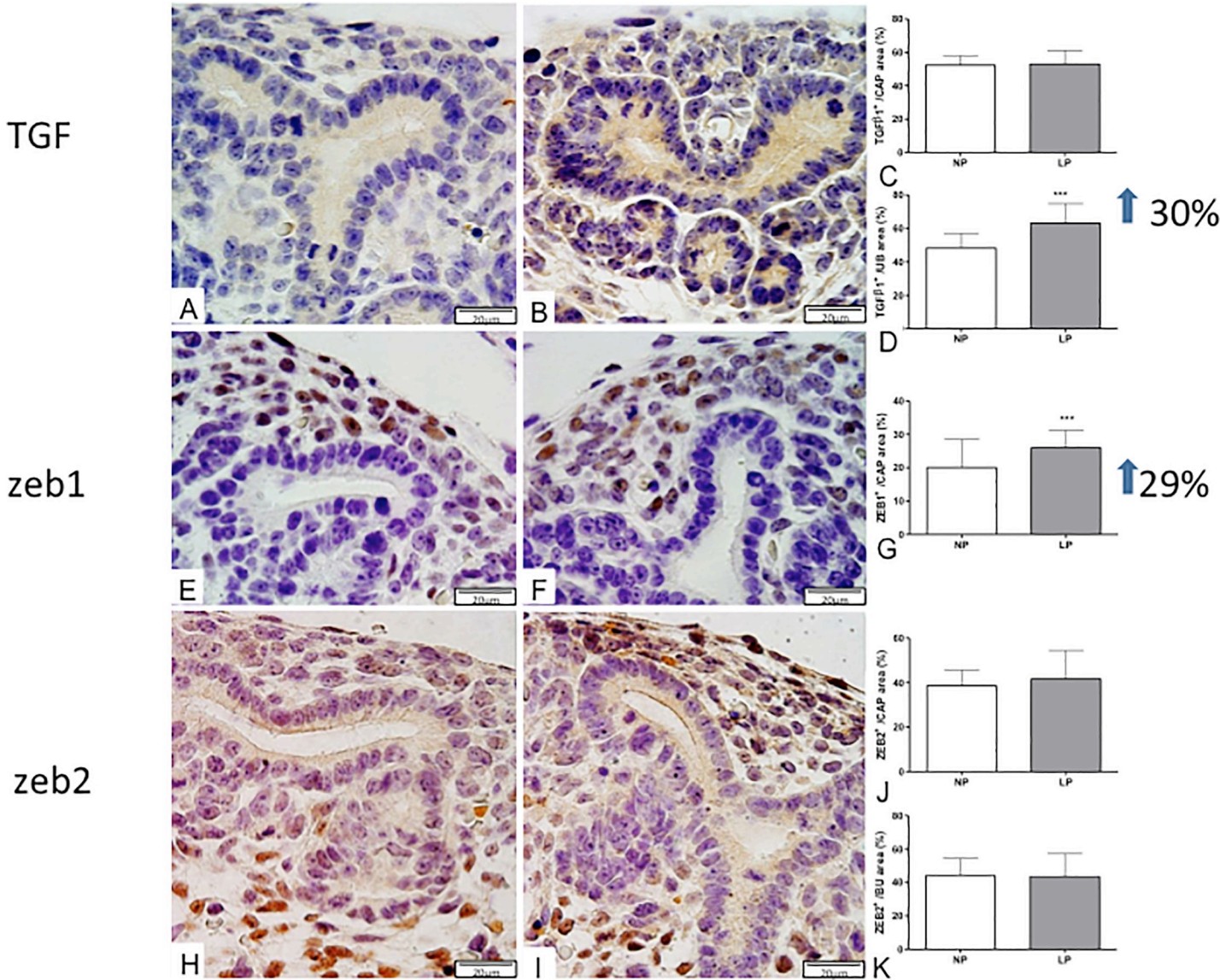

**Fig 8. Immunoperoxidase for TGFβ-1, ZEB1, and ZEB2 in metanephros of 17DG fetus.** The area of TGFβ-1 immunoreactivity in LP (B) compared to NP offspring (A) was not different in the CAP (C) but was significantly enhanced in UB (D). ZEB1 was detected in the nuclei of CAP and other mesenchymal cells and, the LP (E) CAP occupied a more extensive area (G) than in NP (F) offspring. The ZEB2 labeled area was not different from both CAP (J) and UB (K). ***P<0.0001.

signaling path, expressed in both the UB and CM, is required both for nephron progenitor cell renewal and differentiation, being essential for the formation of nephrons during embryogenesis [55]. The evolutionarily conserved Wnt9b/β-catenin pathway plays a critical role in developing organs, tissues, and injury repair in pluricellular organisms. A study demonstrated that c-Myc is a transcriptional target of β-catenin, regulating the proliferation and differentiation of renal tubular epithelium [56]. The expression of β-catenin at the gene and protein level increased during the studied periods of renal development in the 17-DG LP fetus. Pan et al. reported that Myc cooperates with β-catenin to promote the renewal of nephron progenitor cells [10]. Here compared to age-matched NP offspring, LP showed lower c-Myc expression. Therefore, these animals may have a lower reserve of renewable cells necessary for proliferation and survival and may reflect the smaller number of nephrons in the LP model. Moreover,

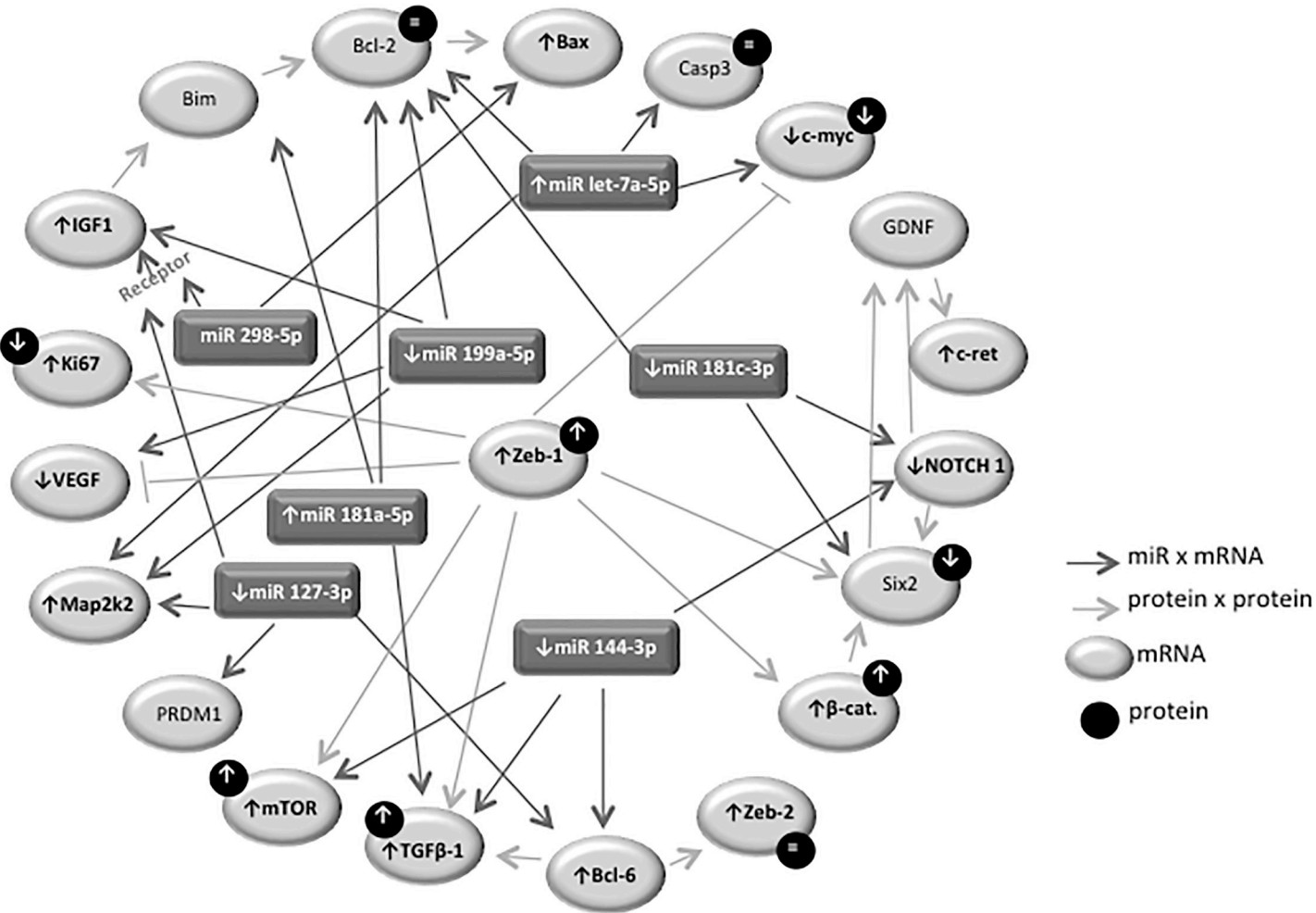

**Fig 9. Deregulated miRNA-mRNA-protein pathways in metanephros of 17DG fetus from maternal restricted-protein intake.**

Wnt/β-catenin and Notch signals pathways may coordinate the regulation of Six-2 expression and are involved in the downregulation of Six-2 expression in nephron progenitor cells. Studies have demonstrated that low levels of β-catenin might be required to maintain Six-2 expression and CM progenitor cells in the undifferentiated state; moreover, elevated levels of β-catenin determine nephron progenitor cell fate [57, 58].Thus, we hypothesize that reduced c-Myc and Notch signaling, accompanied with increased β-catenin expression, reduced Six-2 expression by 28% in 17-DG LP offspring and correlated with early CM cell differentiation and reduced stem cell and nephron number in adulthood. Moreover, our data may sustain that, in LP offspring MM cells, the increased Let-7a-5p and β-catenin expression and reduced Notch signal may modulate c-Myc, Six-2, and Ki-67 expression, leading to a reduction in self-renewal of progenitor cells. The depletion of the remaining CM progenitor cells leads to a reduction in nephron numbers and development of arterial hypertension and renal disorders in adulthood (Fig 10).

Consistent with that of Boivin et al., our results indicate that increased CM β-catenin disrupts UB growth and nephrogenesis [59]. Studies have shown that growth factor glial-derived neurotrophic factor (GDNF), a crucial regulator of UB growth, signals through the c-Ret tyrosine kinase receptor and Gfra1 co-receptor [60, 61]. In 17-DG LP offspring, a significant

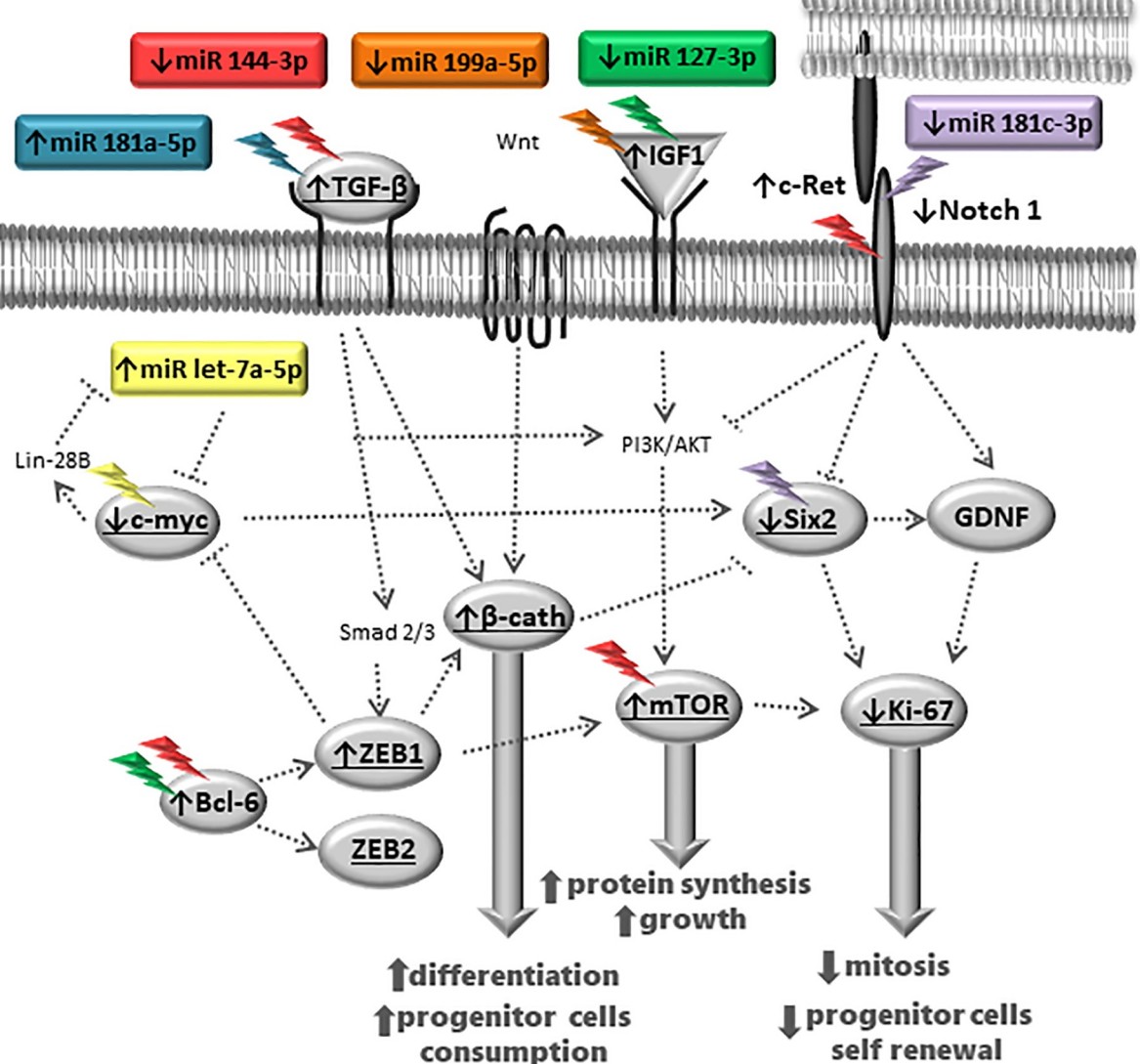

**Fig 10. The picture depicted a schematic representation of the biological response disorders in metanephros of 17DG fetus from maternal restricted-protein intake.**

increase in c-Ret receptor coding mRNA would theoretically lead to a rise in UB growth. However, in this study, GDNF expression was unchanged, suggesting that despite an increase in c-Ret mRNA, UB branching was reduced. Previously, we observed a reduction of 28.3% in ureteric bud branches after 14.5 days of gestational protein restriction [4], which could be associated with a 28% reduction in Six-2 labeling of MM cells, despite no change in GDNF transcription. β-catenin probably interacts with the c-ret receptor and is transported to the UB cell's nucleus, promoting TGFβ-1 expression in epithelial cells, inhibiting UB branching and causing premature differentiation of CM progenitor cell, as seen in 17-DG LP offspring [62–64]. Therefore, GDNF may not be essential in mediating mesenchymal signals to the ureteric bud; however, the mechanism remains to be elucidated. Indeed, we have demonstrated that MM from 17-DG LP offspring showed a specific increase in Let-7 miRNA expression, resulting in significantly impaired kidney development, thereby confirming the modulatory role of these genes in the developmental timing of nephrogenesis [65].

In initial insulin-like growth factor (IGF) studies, predominant roles of IGF-1 and -2 in fetal growth were elucidated by abundant, but mostly indirect, evidence. IGFs were found to act as proliferation and differentiation factors in cultured fetal cells and preimplantation embryos. Moreover, IGFs were found to be secreted by cultured fetal cells and explants *in vitro* [66]. Growth factors, including IGF, can cause a partial or full epithelial–mesenchymal transition. The activation of IGF pathways results in the upregulation of EMT by inducing ZEB1 expression [67]. Although several candidate growth factors are involved in kidney development, whether they are involved in nephrogenesis is unknown. Different growth factors may be needed at different times. Some growth factors may be redundant in this context. During embryonic development, sequential rounds of EMT and MET are needed to differentiate specialized cell types and create a three-dimensional structure. In this study, mesenchymal–epithelial interconvertibility was found to maintain cell plasticity, suggesting the presence of a highly inducible system in LP conditions for the embryo. The expression of the Let-7 miRNA family has been extensively studied in various fetal tissues. The increase in Let-7 miRNA expression is related to reduced proliferation and early increase of MM cell differentiation, and consequently, decreased nephron numbers [27, 15, 68–71]. Higher Let-7 expression has been demonstrated in higher organisms during the last phase of cerebral embryogenesis in rodents [72, 73]. Nagalakshmi et al. revealed that Let-7 miRNA expression changed UB epithelial cell fate from precursor to differentiated state [71]. By contrast, Yermalovich et al. demonstrated that the overexpression of Lin28b, an RNA-binding protein, is associated with suppressive Let-7 miRNAs. Although lin28 and Let-7 are known regulators of ontogenic timing in invertebrates, the role of these in mammalian organ development is not understood [65]. In this study, the increase in Let-7a-5p miRNA expression in the LP fetus could be associated with reduced CM cell proliferation, compromising nephrogenesis relative to the NP group. Thus, we hypothesize that the CM cell proliferation suppression and early cessation of nephrogenesis caused by increased Let-7 miRNA may occur directly or indirectly through the transiently reduced expression of Lin28b in 17-DG LP. This effect may significantly impair kidney development in 17-DG LP, confirming that this gene regulates developmental timing during nephrogenesis. In this study, increased Let-7a-5p miRNA expression coincides with a decrease in c-Myc expression. Myc is involved in proliferation, growth, apoptosis, and cell differentiation during renal organogenesis [74–76]. In the LP 17-DG offspring, MM c-Myc gene expression was reduced, and the area of CM c-Myc immunoreactivity was 14% smaller, when compared to that in the NP offspring. Simultaneously, a 14% decrease in CM cell number was observed to decrease 48% Ki-67 immunoreactivity in LP relative to the NP offspring. Consistently, studies have shown that c-Myc plays an important role in the final phase of UB branching and in stimulating CM progenitor cell proliferation [74].

Let-7 is strongly expressed at late stages of cell differentiation; however, it is expressed at a reduced level in stem cells, maintaining them in an undifferentiated state. However, in this study, strongly expression of Let-7a-5p miRNA in the CM downregulated c-Myc expression, thereby reducing progenitor cell proliferation and early cell differentiation in the LP 17-DG offspring (Fig 10). Studies on the kidneys of c-Myc transgenic mice revealed a simultaneous decrease in c-Myc and S-ix-2 immunopositive CM cells associated with reduced stem cell proliferation [74]. This study shows a significant (28%) reduction in Six-2 positive cells, a specific renal stem cell marker, like the decrease observed in nephron numbers, in the CM of the 17-DG LP offspring compared to that in the NP offspring.

In 2009, Fogelgren et al. demonstrated that Six-2 gene expression is reduced during fetal ontogenesis when associated with decreased nephron numbers, hypertension, and chronic renal failure [77]. Thus, reduced Six-2 gene expression in CM progenitor cells indicates suppression of signal-induced differentiation during renal development in the 17-DG LP

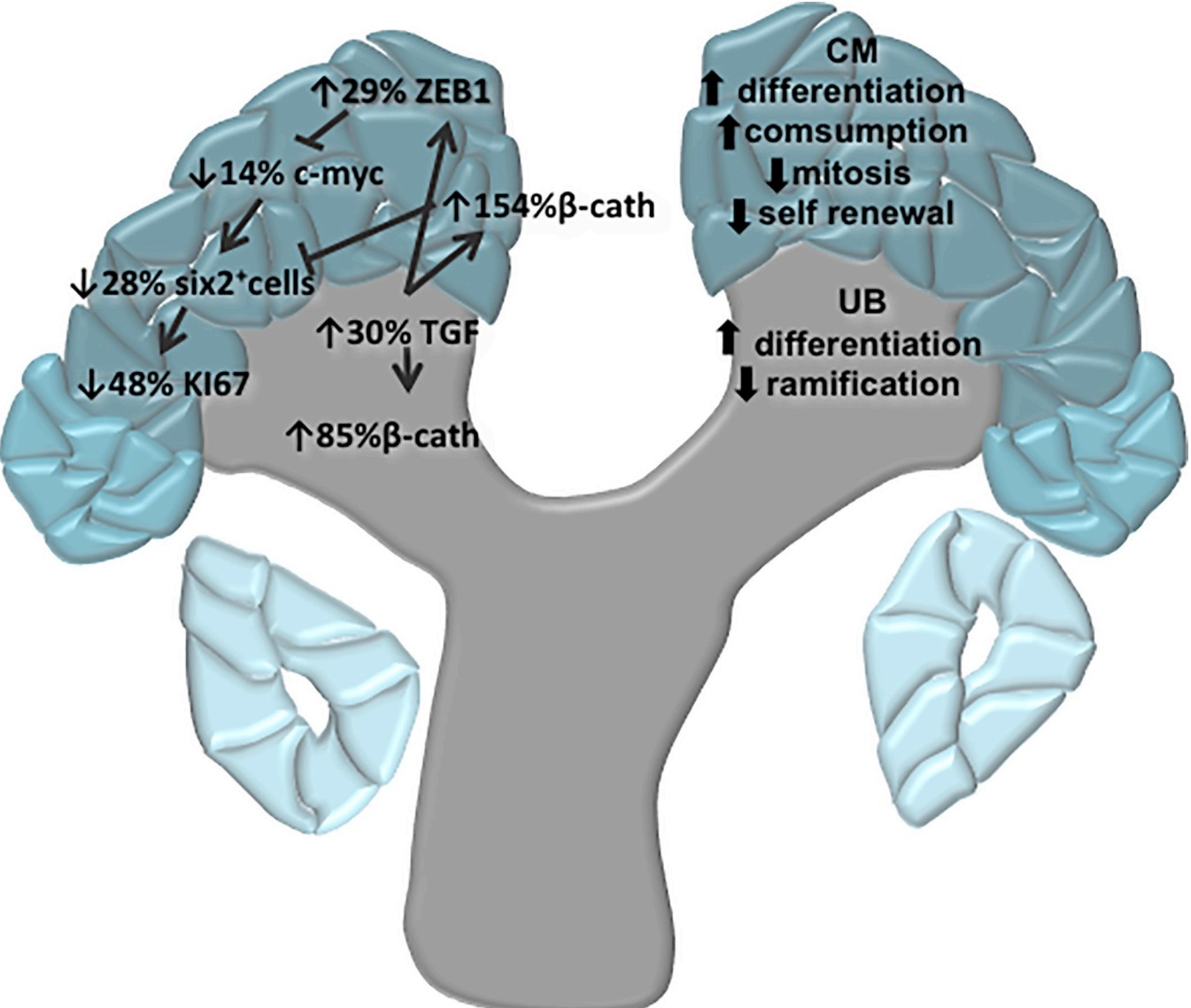

**Fig 11. The picture depicted a schematic representation of supposed factors that evolved in a 28% reduction of the CM stem cells and nephron number in maternal restricted protein intake.**

offspring. Nevertheless, redundancy should be used with caution—subtle defects in nephrogenesis may become evident with more detailed analysis or under different conditions. IGF1 mRNA levels were highest during the initial period of metanephric development, with transcripts being detected throughout the MM, whereas their levels declined during further development. However, during kidney embryogenesis, a delicate balance between nephron facilitating growth factors (IGF1) and inhibitory growth factors (TGFβ-1) regulates UB branching.

## Conclusion

Although several authors have studied nephrogenesis [34, 37, 78], little is known about the mechanisms that determine nephron numbers. This study demonstrates that many MM

progenitor cell miRNAs, mRNAs, and proteins are altered in the 17-DG LP offspring, which leads to reduced proliferation and early cell differentiation (Fig 11).

This delicate balance between nephron progenitor renewal and differentiation is essential for kidney development, because failure to achieve adequate numbers of nephrons is a risk factor for chronic renal disorder.

## Supporting information

**S1 File.**
(DOCX)

## Acknowledgments

We thank the access to equipment and assistance provided by the National Institute of Science and Technology on Photonics Applied to Cell Biology (INFABIC) at the State University of Campinas; Data analysis was partial and generously performed in collaboration with Tao Chen, Ph.D. from the Division of Genetic and Molecular Toxicological, National Center for Toxicological Research, Jefferson, AR, USA.

## Author Contributions

**Conceptualization:** José Antônio Rocha Gontijo, Patrícia Aline Boer.

**Data curation:** Letícia de Barros Sene, Wellerson Rodrigo Scarano, José Antônio Rocha Gontijo, Patrícia Aline Boer.

**Formal analysis:** Letícia de Barros Sene, Wellerson Rodrigo Scarano, José Antônio Rocha Gontijo, Patrícia Aline Boer.

**Funding acquisition:** José Antônio Rocha Gontijo, Patrícia Aline Boer.

**Investigation:** Letícia de Barros Sene, Adriana Zapparoli, Patrícia Aline Boer.

**Methodology:** Letícia de Barros Sene, Wellerson Rodrigo Scarano, Adriana Zapparoli, Patrícia Aline Boer.

**Project administration:** José Antônio Rocha Gontijo, Patrícia Aline Boer.

**Supervision:** Wellerson Rodrigo Scarano, José Antônio Rocha Gontijo, Patrícia Aline Boer.

**Validation:** Letícia de Barros Sene, Wellerson Rodrigo Scarano, Adriana Zapparoli, José Antônio Rocha Gontijo, Patrícia Aline Boer.

**Visualization:** José Antônio Rocha Gontijo, Patrícia Aline Boer.

**Writing – original draft:** Letícia de Barros Sene.

**Writing – review & editing:** José Antônio Rocha Gontijo, Patrícia Aline Boer.

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
