## [Decision Letter · Decision Letter 0]

14 Jul 2020

PONE-D-20-15837

The gestational low-protein intake impact in microRNA expression of the kidney progenitor cells in male offspring fetuses

PLOS ONE

Dear Dr. Gontijo,

Thank you for submitting your manuscript to PLOS ONE. After careful consideration, we feel that it has merit but does not fully meet PLOS ONE’s publication criteria as it currently stands. Therefore, we invite you to submit a revised version of the manuscript that addresses the points raised during the review process, especially those raised by reviewer 2.

We look forward to receiving your revised manuscript.

Kind regards,

Emmanuel A Burdmann

Academic Editor

PLOS ONE

Journal Requirements:

2. Please include further information regarding your in vivo study, per our guidelines (http://journals.plos.org/plosone/s/submission-guidelines#loc-animal-research). Specifically, please provide details regarding:

 1) Animal health monitoring, including:

     -frequency of monitoring, and

     -monitoring criteria

 2) the method of euthanasia for the rats, and

 3) the source of the mice

3. Please provide the missing information for Anti-mTOR in Table 1.

4. We note that Figures 9 and 10 appear to be similar to several general textbook mTOR diagrams. Because many pathway diagrams look similar even if they are not derived from each other, please verify that these figures are your own work and were not derived in part or full from copyrighted images. If they were derived from copyrighted images, you will need to get permissions from the copyright owner. If this is the case, please provide proof that the owner of the content (a) has given you written permission to use it, and (b) has approved of the CC BY license being applied to their content. You may have the following form completed by the owner as proof: https://journals.plos.org/plosone/s/file?id=7c09/content-permission-form.pdf. Alternatively, you may electronically request permissions electronically from the copyright owner and send us proof of approval, as long as the approval clearly shows that the owner has approved of the CC BY license being applied to their content. Please see https://journals.plos.org/plosone/s/licenses-and-copyright for more information.

5. We suggest you thoroughly copyedit your manuscript for language usage, spelling, and grammar. If you do not know anyone who can help you do this, you may wish to consider employing a professional scientific editing service.  

6. Your ethics statement must appear in the Methods section of your manuscript. If your ethics statement is written in any section besides the Methods, please move it to the Methods section and delete it from any other section. Please also ensure that your ethics statement is included in your manuscript, as the ethics section of your online submission will not be published alongside your manuscript.

Reviewers' comments:

Reviewer's Responses to Questions

**Comments to the Author**

1. Is the manuscript technically sound, and do the data support the conclusions?

Reviewer #1: Yes

Reviewer #2: Partly

2. Has the statistical analysis been performed appropriately and rigorously? 

Reviewer #1: Yes

Reviewer #2: Yes

3. Have the authors made all data underlying the findings in their manuscript fully available?

Reviewer #1: Yes

Reviewer #2: Yes

4. Is the manuscript presented in an intelligible fashion and written in standard English?

Reviewer #1: Yes

Reviewer #2: No

5. Review Comments to the Author

Reviewer #1: LB Sene et al. developed an interesting study to evaluate mechanisms responsible for renal structural changes in 17-day-old fetuses of pregnant rats fed a low-protein diet or a diet with normal protein content. The study was very well planned and had as main objective to evaluate micro RNAs and gene and protein expression of several factors. The authors observed relevant differences between the experimental and control groups. These results allows to understand some of the mechanisms responsible for the smaller number of nephrons observed in the offspring of rats subjected to malnutrition during pregnancy.

The manuscript needs a detailed correction of spelling and writing errors. As examples, I would like to draw your attention to some of the following necessary corrections.

Page 10, line 9: replace graph with figure.

Page 12, line four from bottom: replace which with with.

Page 12, line three from bottom: replace could by may

Page 13, line 4: replace "also, here, was demonstrated" by was observed.

These are just a few of the many errors in the manuscript.

In the legend of figure 1, explain the meaning of U6.

The legends of tables 1 and 2 are switched.

It would be interesting to mention some limitations of the study.

Reviewer #2: The manuscript by Sene et al analyzes the impact of maternal protein restriction on molecular aspects of renal development in mice, and reveals that such a restriction modifies the miRNA, mRNA and protein expression scenarios associated with proliferation, apoptosis and differentiation. This is a relevant and timely field of investigation and, in general, the study was adequately performed. The manuscript carries language problems, however, since the English quality is not good, particularly in the discussion. Such problems include grammatical mistakes, sentences that do not allow appropriate understanding (examples shown below in comment #3) and flaws in sentence structure that make the reading sometimes difficult. In this context, the paper should be assessed and have some portions rewritten by a native English speaker. In addition, a number of points should be addressed, clarified or modified before further evaluation. Additional analyses should be also performed for adequate interpretation of some of the data and to allow appropriate conclusions. These points are outlined in my comments below.

1. The authors have carried out the miRNA expression analyses based on statistically significant differences between the LP and NP groups, but have not established a fold-change cutoff for such analyses. They included, however, blue and red dashed lines to establish fold-change upregulation and downregulation thresholds in Figure 1. Some studies use a 2-fold-change cutoff, other studies 1.5 (less often), but such criteria are arbitrary. It is important, however, that the authors define whether they used or not a fold-change cutoff. If they did, please be clear about it and justify the decision. If they did not, please justify why they decided not to use any filtering criteria and explain what and how they defined the fold-change values associated with the blue and red up and down dashed lines.

2. Please also address the fold-change issue to the mRNA context in Figure 2.

3. Examples of inappropriate/unclear sentences:

... predicted gene expression patterns in the 17-days LP (17-DG) fetal kidney to elucidate the molecular pathways and differentiation renal cell proliferation.

Prior studies have shown that during kidney development, the miRNAs underexpression MM progenitor cells results in a premature reduction of cell proliferation and ...

In the current study, increased expression of miR-181a-5p in 17-DG LP relative to age-matched NP offspring; also, here, was demonstrated a 2-fold enhanced Bax/Bcl2 mRNA ratio …

… they also showed that enhanced miR-144 expression suppresses renal carcinoma proliferation and decreasing the G2/M phase cells ratio.

The let-7 miRNA family expression has been extensively studied in several fetal tissues and, priority is related with reduced proliferation and induced cell differentiation.

It has been shown in higher organisms, enhanced let-7 levels during embryogenesis (Schulman et al., 2005), and let-7a mature form is up-regulated during the developmental mouse brain.

How is it known that Six2 regulates transcription of GDNF (Brodbeck et al., 2004), thus, the reduction of 28% in the cells positive for Six2 could affect, in the same proportion the GDNF expression which in turn, would act in …

4. The study associates increased expression of miR 181a-5p in 17-DG LP offspring with increased Bax/Bcl2 mRNA ratio to explain increased apoptosis activity, despite no change in caspase mRNA expression. Since Bax overexpression has been shown to induce caspase-independent apoptosis and cordycepol C has been shown to induce caspase-independent apoptosis in HepG2 cells through a Bax-mediated mitochondrial pathway, I suggest to investigate caspase-independent mechanisms of apoptosis in 17-DG LP fetal kidneys.

5. Please discuss potential mechanisms relating reduced expression of miR-144-3p in 17-DG LP offspring CAP and decreased cell proliferation.

6. As pointed out by the authors, increased activity of mTOR led to nephron number reduction in fetal kidneys while hemizygous removal of mTOR also diminishes nephron population. Is there a narrow range of mTOR activity during nephrogenesis that appropriately regulates nephron number? Please discuss this issue and apply this discussion to analyze the current model.

7. Please clarify the sentence “Increased mRNA accompanied the reduction of miR 127-3p in 17-DG LP offspring for Ki67 associated with an increase of Bcl-6 in CM”.

8. Because Ki67 gene expression is increased and Ki-67 immunoreactivity is decreased in LP 17-DG metanephros, the authors state that gestational undernutrition promotes differentiation in detriment of proliferation. If so, this is a post-transcriptional mechanism. Please discuss how that may occur, cite other models in which a similar process occurred, and clarify the association with Zeb2 expression.

9. Please clarify the sentence “On the other hand, Yermalovich et al. (2019) have demonstrated the overexpression of Lin28b, an RNA-binding protein, is associated with suppressive let-7 miRNA expression elongated nephrogenesis, via the let-7 miRNAs upregulation.”

10. Since the authors hypothesize that overexpression of let-7 miRNAs, through a transient reduction of LIN28B, might decrease nephrogenesis and consequently the nephron number potentially via upregulation of Igf2, I recommend them to check Igf2 expression in the current LP model.

11. The authors state that “the current study established that the let-7 family of miRNAs promotes MYC expression through transcriptionally induced let-7 repressor, LIN28 enhancement and posttranscriptional expressed LIN28 RNA binding-protein, promoting downregulation upon LP kidney cells differentiation”, however there is no generated data on let-7 repressor and/or Lin28 that support this conclusion.

12. Notch signaling has been shown to promote nephrogenesis by downregulating Six2. In the current study, the authors show decreased Notch1 expression but unchanged Six2 expression and reduction of Six-2 positive cells in LP offspring metanephros. In this scenario, the statement “the increased let-7a-5p and β-catenin expression and reduced Notch signal modulate the c-myc, six2, and KI-67, leading to reduction of progenitor cells self-renewal in LP metanephros” should not be presented as a conclusion but instead as “suggests that the increased let-7a-5p and β-catenin expression and reduced Notch signal may modulate the c-myc, six2, and KI-67, leading to …”.

13. Given that c-Ret receptor tyrosine kinase is a major inducer of UB branching, the increase in c-Ret expression is expected to increase UB branching even if the expression of GDNF is unchanged. Moreover, despite the reduction of Six-2 positive cells, GDNF expression did not change. In this scenario, the current discussion does not seem to appropriately support the observed 28.3% reduction in UB branching.

6. PLOS authors have the option to publish the peer review history of their article (what does this mean?). If published, this will include your full peer review and any attached files.

Reviewer #1: No

Reviewer #2: No

---

## [Author Response · Author response to Decision Letter 0]

6 Nov 2020

Response to Reviewers

PONE-D-20-15837

The gestational low-protein intake impact in microRNA expression of the kidney progenitor cells in male offspring fetuses

PLOS ONE

Dear Editor:

We are attaching the revised version and response to the Reviewers of the Manuscript PONE-D-20-15837. The gestational low-protein intake impact in microRNA expression of the kidney progenitor cells in male offspring fetuses submitted by Sene et al. resubmitted as a full paper. I have read and have abided by the statement of ethical standards for manuscripts submitted to the Plos One and the other information that all authors have approved the final article. This manuscript has not been submitted or published in any other journal. At first, thank you very much for the reviewer's comments, suggestions, and criticisms. Practically all sections of the manuscript were rewritten entirely, experiments are redone, the number of experiments has been increased, and many reviewer suggestions were now included in that new version. The grammatical and typographical errors have been corrected. The Introduction, Material, and Method and Discussion sections of the manuscript were revised and completely rewritten to include reviewers' suggestions and comments.

Hopefully, this edited manuscript version could now be better considered for publication by this prestigious Journal.

Sincerely yours,

José AR Gontijo, MD, 

Internal Medicine Department, School of Medicine, 

Campinas State University, Campinas, SP, Brazil. 

E-mail: jgontijo@unicamp.br

A. Journal Requirements:

1. Please include further information regarding your in vivo study, per our guidelines (http://journals.plos.org/plosone/s/submission-guidelines#loc-animal-research). 

Specifically, please provide details regarding:

 1) Animal health monitoring, including:

 -frequency of monitoring, and

 -monitoring criteria

 2) the method of euthanasia for the rats, and

 3) the source of the mice

2. Please provide the missing information for Anti-mTOR in Table 1.

3. Your ethics statement must appear in the Methods section of your manuscript. If your ethics statement is written in any section besides the Methods, please move it to the Methods section and delete it from any other section. Please also ensure that your ethics statement is included in your manuscript, as the ethics section of your online submission will not be published alongside your manuscript.

B. Review Comments to the Author

Reviewer #1: LB Sene et al. developed an interesting study to evaluate mechanisms responsible for renal structural changes in 17-day-old fetuses of pregnant rats fed a low-protein diet or a diet with normal protein content. The study was very well planned and had as main objective to evaluate micro RNAs and gene and protein expression of several factors. The authors observed relevant differences between the experimental and control groups. These results allows to understand some of the mechanisms responsible for the smaller number of nephrons observed in the offspring of rats subjected to malnutrition during pregnancy. The manuscript needs a detailed correction of spelling and writing errors. As examples, I would like to draw your attention to some of the following necessary corrections.

Response to Reviewer 1 

REF.: PONE-D-20-15837

At first, thank you very much for your criticisms. We have greatly appreciated your comments and suggestions. Practically all sections of the manuscript were rewritten entirely, experiments are redone, the number of experiments has been increased, and many reviewer suggestions were now included in that new version. The grammatical and typographical errors have been corrected. The Introduction, Material, and Method and Discussion sections of the manuscript were revised and completely rewritten to include the suggestions and comments of reviewer 1.

Major comments

Response from the authors:

1. Page 10, line 9: replace graph with figure. 

R.: Regarding the suggestion brought up by Reviewer 1, the word was replaced.

2. Page 12, line four from bottom: replace which with with

R.: Regarding the suggestion brought up by Reviewer 1, the word was replaced.

3. Page 12, line three from bottom: replace could by may OK

R.: Regarding the suggestion brought up by Reviewer 1, the word was replaced.

4. Page 13, line 4: replace "also, here, was demonstrated" by was observed. OK

These are just a few of the many errors in the manuscript.

R.: Regarding the suggestion brought up by Reviewer 1, the phrase was rewitten.

5. In the legend of figure 1, explain the meaning of U6. OK

R.: Regarding the suggestion brought up by Reviewer 1, the meaning of U6 was included in the legend of Figure 1, to read… Expression of miRNAs in the metanephros from the 17th day LP fetus compared to their expression level in the control group. Reference genes U6 and U87, protein complexes composed of small nuclear RNAs (snRNAs), were used to normalize the expression of each miRNA. Data are expressed as fold change (mean ± SD, n = 4) concerning the control group. * p≤0.05: statistical significance versus NP.

And also in Methods, to read: … Based on stability analysis, the U6 snRNA, and U87 scaRNA were used as a reference gene.

Recently the improvements in high-throughput gene expression analysis have led to numerous non-protein-coding RNA (npcRNA) molecules. 

Non-protein-coding RNAs are untranslated RNA molecules frequently playing regulatory roles in different developmental and cellular processes. U87 scaRNA, and U6 snRNA) possess significantly higher expression stability than the best protein-coding housekeeping RNAs. Based on our stability analysis are recommended the inclusion of U6 snRNA, and U87 scaRNA in the minimal set of reference candidates to be evaluated for normalization in any given npcRNA transcriptome analysis.

6. The legends of tables 1 and 2 are switched. 

R.: Regarding the suggestion brought up by Reviewer 1, the legends of Tables 1 and 2 were switched.

Review Comments to the Author

Reviewer #2: The manuscript by Sene et al analyzes the impact of maternal protein restriction on molecular aspects of renal development in mice, and reveals that such a restriction modifies the miRNA, mRNA and protein expression scenarios associated with proliferation, apoptosis and differentiation. This is a relevant and timely field of investigation and, in general, the study was adequately performed. The manuscript carries language problems, however, since the English quality is not good, particularly in the discussion. Such problems include grammatical mistakes, sentences that do not allow appropriate understanding (examples shown below in comment #3) and flaws in sentence structure that make the reading sometimes difficult. In this context, the paper should be assessed and have some portions rewritten by a native English speaker. In addition, a number of points should be addressed, clarified or modified before further evaluation. Additional analyses should be also performed for adequate interpretation of some of the data and to allow appropriate conclusions. These points are outlined in my comments below.

Response to Reviewer 2 

REF.: PONE-D-20-15837

At first, thank you very much for your criticisms. We have greatly appreciated your comments and suggestions. Practically all sections of the manuscript were rewritten entirely, experiments are redone, the number of experiments has been increased, and many reviewer suggestions were now included in that new version. The grammatical and typographical errors have been corrected. The Introduction, Material, and Method and Discussion sections of the manuscript were revised and completely rewritten to include the suggestions and comments of reviewer 1.

Major comments

Response from authors:

1. The authors have carried out the miRNA expression analyses based on statistically significant differences between the LP and NP groups, but have not established a fold-change cutoff for such analyses. They included, however, blue and red dashed lines to establish fold-change upregulation and downregulation thresholds in Figure 1. Some studies use a 2-fold-change cutoff, other studies 1.5 (less often), but such criteria are arbitrary. It is important, however, that the authors define whether they used or not a fold-change cutoff. If they did, please be clear about it and justify the decision. If they did not, please justify why they decided not to use any filtering criteria and explain what and how they defined the fold-change values associated with the blue and red up and down dashed lines.

R.: Bearing in mind the question raised by Reviewer 2, studies have used arbitrary criteria to define the upper and lower cutoff points related to gene expression. As this value is defined arbitrarily, the present study's authors established a cutoff point variation of 1.3 fold-change related to control group values. The change of the 1.3 (upwards) and 0.65 (downwards) were defined since the statistical analysis found a P-value to miRs and target mRNAs significant concerning the NP. Therefore, as we have established these cutoff points for miR, we maintained the same to validate the sequencing differences.

2. Please also address the fold-change issue to the mRNA context in Figure 2.

R.: Bearing in mind the question raised by Reviewer 2, studies have used arbitrary criteria to define the upper and lower cutoff points related to gene expression. As this value is defined arbitrarily, the present study's authors established a cutoff point variation of 1.3 fold-change related to control group values. The change of the 1.3 (upwards) and 0.65 (downwards) were defined since the statistical analysis found a P-value to miRs and target mRNAs significant concerning the NP. Therefore, as we have established these cutoff points for miR, we maintained the same to validate the sequencing differences.

3. Examples of inappropriate/unclear sentences:

3.1 ...predicted gene expression patterns in the 17-days LP (17-DG) fetal kidney to elucidate the molecular pathways and differentiation renal cell proliferation.

R.: The sentence was re-written to read: ...The current study evaluated the miRNAs and kidney predicted gene expression patterns to gestational 17-days LP (DG-17) offspring elucidates the molecular pathways involved in the proliferation and differentiation of renal embryonic/fetal cell.

3.2 Prior studies have shown that during kidney development, the miRNAs underexpression MM progenitor cells results in a premature reduction of cell proliferation and ...

R.: The sentence was re-written to read: ...Previous studies have shown that some miRNAs' underexpression in mesenchymal metanephros (MM) progenitor cells, during renal ontogenesis, results in a reduction in the cell proliferation process with early differentiation of these cells and, consequently, decreased the number of nephrons (Ho et al., 2011; Nagalakshmi et al., 2011).

3.3 In the current study, was observed increased expression of miR-181a-5p in 17-DG LP relative to age-matched NP offspring; also, here, was demonstrated a 2-fold enhanced Bax/Bcl2 mRNA ratio …

R.: The sentence was re-written to read: ...The current study showed an increased expression of miR-181a-5p in 17-DG LP relative to age-matched NP offspring. Although caspase mRNA expression was not altered, a 2-fold enhanced Bax/Bcl2 mRNA ratio in LP compared to NP offspring suggests an increased CM apoptosis activity, indicating a post-transcriptional mechanism apoptosis regulation.

3.4 … they also showed that enhanced miR-144 expression suppresses renal carcinoma proliferation and decreasing the G2/M phase of cell cycle.

R.: The sentence was re-written to read: Furthermore, studies of Xiang et al. demonstrated that miR-144 expression was decreased in kidney carcinoma cells and that enhanced miR-144 expression suppresses carcinoma proliferation, reducing the cell cycle's G2/M phase.

3.5 The let-7 miRNA family expression has been extensively studied in several fetal tissues and, priority is related with reduced proliferation and induced cell differentiation.

R.: The sentence was re-written to read: ...Besides, the expression of the let-7 miRNA family has been extensively studied in different fetal tissues. The enhanced let-7 miRNA is related to reduced proliferation and early increase of metanephric mesenchyme cells differentiation and, consequently, decreased the number of nephrons (Ho et al., 2011; Ambros, 2012; Bao et al., 2013; Copley and Eaves, 2013; Meza-Sosa et al., 2014; Nagalakshmi et al., 2015).

3.6 It has been shown in higher organisms, enhanced let-7 levels during embryogenesis (Schulman et al., 2005), and let-7a mature form is up-regulated during the developmental mouse brain.

R.: The sentence was re-written to read: ...Higher let-7 expression has been demonstrated in higher organisms during the last phase of cerebral embryogenesis in rodents (Schulman et al., 2005; Wulczyn et al., 2007).

3.7 How is it known that Six2 regulates transcription of GDNF (Brodbeck et al., 2004), thus, the reduction of 28% in the cells positive for Six2 could affect, in the same proportion the GDNF expression which in turn, would act in …

R.: The sentence was re-written to read: ...As Six2 is known to regulate GDNF transcription (Brodbeck et al., 2004), the 28% reduction in labeling for Six2 cells of the metanephric mesenchyme can affect, in the same proportion GDNF expression, which in turn would act in the decrease of 28.3% of the ramifications of the ureteric bud as previously observed (Mesquita et al., 2010).

4. The study associates increased expression of miR 181a-5p in 17-DG LP offspring with increased Bax/Bcl2 mRNA ratio to explain increased apoptosis activity, despite no change in caspase mRNA expression. Since Bax overexpression has been shown to induce caspase-independent apoptosis and cordycepol C has been shown to induce caspase-independent apoptosis in HepG2 cells through a Bax-mediated mitochondrial pathway, I suggest to investigate caspase-independent mechanisms of apoptosis in 17-DG LP fetal kidneys.

R.: We thanks reviewer 2 for suggestions for the benefit of data interpretation and clarity of Discussion. …We recognize that lack or excess of nutrients is vital during embryo/fetus development stages can have irreversible consequences. Our model shows that males subjected to gestational 6% protein restriction dams showed a ~ 28% reduction in the number of nephrons. On the 17th LP gestational day, both the number of buds in the ureter and the number of nephron-progenitor cells in the metanephrogenic Cap is 28% reduced. In that circumstance (low-protein diet) and depending on the passive exhaustion mechanism, the subpopulation of self-renewable stem cells in the CM can be early depleted. During kidney development, part of primordial cells differentiates and partly remains in divisions maintaining the necessary number of stem cells sufficient to complete the nephrogenesis. We hypothesized that the lack of maternal nutrients limits amino acids' availability for protein synthesis critical to both stem cell mitosis and differentiation of these cells. Additionally, it is plausible to assume that the apoptosis process or/and autophagy mechanisms may be acting on these primordial cells submitted to an insufficient amount of nutrients within the microenvironment formed by the CM's cluster of intensive growths and in differentiation. We also have seen that animals raised to protein restriction even during the breastfeeding period showed an additional 10% reduction in nephrons units; preliminary data from our lab assume that autophagy is also occurring. Thus, as suggest the Reviewer 2, right now, we are performing studies that aim to verify the occurrence of autophagy and involvement of apoptosis path in nephron progenitor cells of male and female metanephric mesenchyme in different intrauterine days during renal development from the offspring of mice submitted to gestational protein restriction, comparatively to their controls different periods of renal ontogenesis.

5. Please discuss potential mechanisms relating reduced expression of miR-144-3p in 17-DG LP offspring CAP and decreased cell proliferation. And, as pointed out by the authors, increased activity of mTOR led to nephron number reduction in fetal kidneys while hemizygous removal of mTOR also diminishes nephron population. Is there a narrow range of mTOR activity during nephrogenesis that appropriately regulates nephron number? Please discuss this issue and apply this discussion to analyze the current model.

R.: Regarding the comments brought up by Reviewer 2, the text was rewritten to read: ...The study of Xiang et al. demonstrated that enhanced miR-144 expression suppresses renal carcinoma proliferation, reducing the cell cycle's G2/M phase. The Xiang group also showed that overexpression of miR-144 inhibits the mTOR gene and protein expression (Xiang et al., 2016). Previously, Nijland and col. demonstrated that an increased mTOR-signaling is crucial for determining the number of nephrons in embryos whose mothers were subjected to a nutrient restriction (Nijland et al., 2007). The mammalian target of rapamycin complex 1 (mTORC1) is known to be essential for embryos development; however, how this complex regulates the balance between growth and autophagy in physiological conditions and environmental stress remains unknown (Gürke et al., 2016). Therefore, it has been suggested that mTOR signaling would undoubtedly be involved in cellular responses in maternal protein underfeeding animals and plays an intricate role in the perception, induction process, and termination of autophagy and response to intracellular nutrient availability (Xiang et al., 2016). By hypothesis, in the present severe protein-restricted study, a reduced expression of miR-144-3p may be associated with remarkably increased mTOR expression, about 139%, and 104% in CM cells and UB, as supposedly a containment mechanism to reduce the lost number of nephrons in the 17-DG LP offspring.

6. Please clarify the sentence “Increased mRNA accompanied the reduction of miR 127-3p in 17-DG LP offspring for Ki67 associated with an increase of Bcl-6 in CM”.

R.: Regarding the comments brought up by Reviewer 2, the text was rewritten to read: …Pan et col. show in liver cells that miR-127 under-expression is related to increased cell proliferation (Pan et al., 2012). Since the present study showed a decreased cell proliferation and significant reduction of cells positively labeled for Ki67 in CM in protein-restricted animals, we may infer that increased Ki67 and Bcl-6 mRNA expression accompanied by reduced miR-127-3p in the 17-DG LP cap could be associated with presumably counter-regulatory mechanisms aimed at maintaining the proliferative process.

7. Because Ki67 gene expression is increased and Ki-67 immunoreactivity is decreased in LP 17-DG metanephros, the authors state that gestational undernutrition promotes differentiation in detriment of proliferation. If so, this is a post-transcriptional mechanism. Please discuss how that may occur, cite other models in which a similar process occurred, and clarify the association with Zeb2 expression.

R.: Regarding the comments brought up by Reviewer 2, the text was rewritten to read: … The present study showed a decreased cell proliferation and significant reduction of cells positively labeled for Ki67 in CM in protein-restricted animals. The present study also demonstrates a reduced nephrogenic area and proliferation activity in LP progeny, confirming Menendez-Castro and col. studies (2013, 2014) in 8.4% protein-restricted progeny. We may infer that increased Ki67 and Bcl-6 mRNA expression accompanied by reduced miR-127-3p in the 17-DG LP cap could be associated with presumably counter-regulatory mechanisms to maintain the proliferative process. Sun et col have demonstrated that overexpression of miR-199a-5p reduces cystic cell proliferation and induces apoptosis, in addition to controlling the cell cycle (Sun et al., 2015). In the current study, expression of miR-199a-5p is reduced in the 17-DG LP cap accompanied by increased transcription of Ki67, a cellular proliferation marker, and Map2k2 is associated with decreased Ki-67 reactivity in LP 17-DG metanephros. This finding suggests that gestational undernutrition promotes differentiation in detriment of proliferation by a post-transcriptional mechanism. Surprisingly, this study showed a repressive role of zinc-finger E-box binding homeobox 1 (ZEB1) expression, a crucial inducer of EMT supposedly to maintain stem cell pluripotency on embryonic stem cell differentiation. It is known β-catenin activates nuclear ZEB1 transcription resulted in ZEB1 expression [34]. One of the best-studied pathways that were already early identified as having EMT inducing capacities during embryologic development is the TGFβ signaling pathway. Indeed, a multitude of TGFβ like ligands is necessary for proper embryonic development. However, it is clear that not all TGFβ mediated effects on EMT depend on ZEB1/2 once knocked cells can still induce the mesenchymal genes fibronectin N-cadherin. However, E-cadherin is no longer downregulated, and actin fibers are even formed (Shirakihara et al., 2007). During renal development, Karner et al. (2011) demonstrated that the Wnt9b/β-catenin signaling path, expressed in both UB and CM, is required both for nephron progenitor cell renewal and differentiation being essential for the formation of nephrons during embryogenesis. The evolutionary conserved Wnt9b/β-catenin pathway plays a critical role in developing organs, tissues, and injury repair in pluricellular organisms. The study demonstrated that c-myc is a transcriptional target of β-catenin, regulating renal tubular epithelium's proliferation and differentiation (Hu and Rosenblum, 2005). The gene and protein expression of β-catenin increases during the studied renal development periods in the 17-DG LP fetus. Recently the Pan group showed that myc cooperates with β-catenin to enhance the renewal of nephron progenitor cells (Pan et al., 2017). Here, the LP compared to age-matched NP offspring showed lower c-myc expression. So we can assume these animals may have a lower reserve of renewing cells necessary for proliferation and survival and may reflect on the smaller number of nephrons in the LP model. Also, Wnt/β-catenin and Notch signals pathways may coordinate that Six2 expression regulation and are involved in the downregulated Six2 expression in nephron progenitor cells. Have been previously demonstrated that a low level of β-catenin might be required for the maintenance of Six2 expression and the CM progenitor cell in the undifferentiated phase and, also that elevated levels of β-catenin levels also determine nephron progenitor cell fate (Cheng, 2003; Cheng et al., 2009). Thus, we may hypothesize that reduced c-myc and Notch signaling accompanied by enhanced β-catenin expression cause 28% reduced Six2 expression in the 17-DG LP offspring related to early CM cell differentiation, reduced stem cells, and nephron number adulthood. Also, our data may sustain that, in LP offspring MM cells, the increased let-7a-5p and β-catenin expression and reduced Notch signal modulate the c-myc and six2, leading to a reduction of progenitor cells self-renewal. By the way, the exhaustion of the remaining CM progenitor cell endowment, in turn, predispose reduced nephron numbers, arterial hypertension development, and renal disorders in adult age (Figure 10). In the current study, confirming Boivin et al. (2015), we may state that increased CM β-catenin might disrupt UB ramifications and nephrogenesis. Prior data observed in our Lab, showed a reduction of 28.3% in ureteric bud branches after 14.5 days of gestational protein restriction (Mesquita et al., 2010). Have been previously known that growth factor glial-derived neurotrophic factor (GDNF) encoding genes encoding, a crucial regulator of UB outgrowth acts via c-Ret tyrosine kinase receptor and Gfra1 co-receptor (Davis et al., 2015). In the 17-DG LP offspring, a significant increase in c-Ret receptor coding mRNA would theoretically lead to a rise in UB ramifications. However, in the present study, the GDNF expression was unchanged, being plausible to suppose that, despite the increase of c-Ret mRNA, the UB branching was reduced. The decrease of 28.3% ureteric bud ramifications previously observed by Mesquita et col, (2010) could be associate with a 28% reduction in labeling for Six2 cells in the MM, despite unchanged GDNF transcription. Besides, we may suppose that β-catenin interacts with the c-ret receptor and is transported to the UB cell nucleus, promoting precociously increasing TGFβ-1 expression in epithelial cells. Inhibiting UB branching and causing premature differentiation of CM progenitor cells (Bridgewater et al., 2008; Marose et al., 2009; Bridgewater et al., 2011), such as seen in 17-DG LP offspring. In this way, perhaps in this circumstance, GDNF is not essential in mediating mesenchymal signals to the ureteric bud; however, its action's exact mechanism remains to be elucidated. Indeed, we show that MM from 17-DG LP offspring showed specific enhanced let-7 miRNA expression, resulting in significantly impaired kidney development, confirming the crucial modulation role of these genes in proper developmental timing of nephrogenesis (Yermalovich et al., 2019). Therefore, we may hypothesize that up-regulated let-7 miRNAs, directly or transient by increased expression of LIN28B, controlling the early cessation of nephrogenesis. In initial IGF studies, the predominant roles of IGF-I and -II in fetal growth were elucidated by abundant but mostly indirect evidence. IGFs were shown to act as proliferation and differentiation factors in cultured fetal cells and preimplantation embryos. They were demonstrated to be secreted by cultured fetal cells and explants in vitro (Agrogiannis et al., 2014). Also, growth factors, including IGF, can cause a partial or full mesenchymal transition of the epithelial cells. Activation of IGF pathways results in EMT's upregulation by inducing ZEB1 expression (Ding et al., 2010). Although several candidate growth factors are involved in kidney development, it is unknown whether they must carry out nephrogenesis. Different growth factors may be needed at different times. Some of the growth factors may also be redundant in this particular context. During embryonic development, sequential rounds of EMT and MET are needed to differentiate specialized cell types and create the three-dimensional structure. In the present case, a mesenchymal-epithelial interconverted ability sustains cell plasticity, suggesting a highly induced phenomenon in the embryonic protein low-nutrition condition. 

8. Please clarify the sentence “On the other hand, Yermalovich et al. (2019) have demonstrated the overexpression of Lin28b, an RNA-binding protein, is associated with suppressive let-7 miRNA expression elongated nephrogenesis, via the let-7 miRNAs upregulation.”

R.: We thanks reviewer 2 for suggestions for the benefit of data interpretation and clarity of Discussion, the text was rewritten to read…On the other hand, Yermalovich et al. (2019) have demonstrated that the overexpression of Lin28b, an RNA-binding protein, is associated with suppressive let-7 miRNAs. In contrast, kidney-specific loss of Lin28b impairs renal development. While the lin28 and let-7 genes are well-established regulators of ontogenic timing in invertebrates, the role of these in mammalian organ development was not fully understood. In the current study, the enhanced let-7a-5p miRNA expression in LP offspring could be associated with reducing CM cell proliferation, compromising the whole nephrogenesis relative to the NP group. The unprecedented CM cell proliferation suppression caused by increased let-7 miRNAs may occur directly or via the transient overexpression of Lin28b in the 17-DG LP.

9. Since the authors hypothesize that overexpression of let-7 miRNAs, through a transient reduction of LIN28B, might decrease nephrogenesis and consequently the nephron number potentially via upregulation of Igf2, I recommend them to check Igf2 expression in the current LP model.

R.: Regarding the comments brought up by Reviewer 2, suggesting includes studying expression and protein content for Igf2 in the gestational protein restriction model. Most studies reported a higher abundance of Igf2 mRNA in fetal tissues than adult tissues. This raised the suggestion that IGF-II is the IGF that mediates growth and differentiation in developing fetal tissues. However, at this time, we do not have inputs and fetal material available for these assessments. As we are continuing to evaluate this topic through studies with embryonic material, we are committed to assessing the expression of Igf2 in renal embryonic tissue in the future.

10. The authors state that “the current study established that the let-7 family of miRNAs promotes MYC expression through transcriptionally induced let-7 repressor, LIN28 enhancement and posttranscriptional expressed LIN28 RNA binding-protein, promoting downregulation upon LP kidney cells differentiation”, however there is no generated data on let-7 repressor and/or Lin28 that support this conclusion.

R.: Regarding the divergences between results obtained from gene expression and what would be expected, taking into account the miR expression that targets these mRNAs, we could explain by the fact that the miR binding to the target mRNA is in two ways: complete or partial, so if complete, there is the degradation of the target messenger, but when partial, it is not degraded (Bartel, 2013). These data may also differ since each miRNA can regulate the expression of several target mRNAs, and the presentation of each mRNA can be handled by several miRNAs (Enright et al., 2003; Van Rooiji et al., 2008).

11. Notch signaling has been shown to promote nephrogenesis by downregulating Six2. In the current study, the authors show decreased Notch1 expression but unchanged Six2 expression and reduction of Six-2 positive cells in LP offspring metanephros. In this scenario, the statement “the increased let-7a-5p and β-catenin expression and reduced Notch signal may modulate the c-myc, six2, and KI-67, leading to, leading to reduction of progenitor cells self-renewal in LP metanephros” should not be presented as a conclusion but instead as “suggests that the increased let-7a-5p and β-catenin expression and reduced Notch signal may modulate the c-myc, six2, and KI-67, leading to …”.

R.: Regarding the comments brought up by Reviewer 2, the text was rewritten to read:…Thus, we may hypothesize that reduced c-myc and Notch signaling accompanied by enhanced β-catenin expression cause 28% reduced Six2 expression in the 17-DG LP offspring related to early CM cell differentiation, reduced stem cells, and nephron number adulthood. Also, our data may sustain that, in LP offspring MM cells, suggests that the increased let-7a-5p and β-catenin expression and reduced Notch signal may modulate the c-myc, six2, and KI-67, leading to a reduction of progenitor cells self-renewal. By the way, the exhaustion of the remaining CM progenitor cell endowment, in turn, predispose reduced nephron numbers, arterial hypertension development, and renal disorders in adult age (Figure 10).

12. Given that c-Ret receptor tyrosine kinase is a major inducer of UB branching, the increase in c-Ret expression is expected to increase UB branching even if the expression of GDNF is unchanged. Moreover, despite the reduction of Six-2 positive cells, GDNF expression did not change. In this scenario, the current discussion does not seem to appropriately support the observed 28.3% reduction in UB branching.

R.: Regarding the comments brought up by Reviewer 2 for suggestions for the benefit of data interpretation and clarity of Discussion, the text was rewritten to read…However, in the present study, the GDNF expression was unchanged, being plausible to suppose that, despite the increase of c-Ret mRNA, the UB branching was reduced. Prior data observed in our Lab, showed a reduction of 28.3% in ureteric bud branches after 14.5 days of gestational protein restriction (Mesquita et al., 2010) could be associate with a 28% reduction in labeling for Six2 cells in the MM, despite unchanged GDNF transcription. Besides, we may suppose that β-catenin interacts with the c-ret receptor and is transported to the UB cell nucleus, promoting precociously increasing TGFβ-1 expression in epithelial cells. Inhibiting UB branching and causing premature differentiation of CM progenitor cells (Bridgewater et al., 2008; Marose et al., 2009; Bridgewater et al., 2011), such as seen in 17-DG LP offspring. In this way, perhaps in this circumstance, GDNF is not essential in mediating mesenchymal signals to the ureteric bud; however, its action's exact mechanism remains to be elucidated. Indeed, we show that MM from 17-DG LP offspring showed specific enhanced let-7 miRNA expression, resulting in significantly impaired kidney development, confirming the crucial modulation role of these genes in proper developmental timing of nephrogenesis (Yermalovich et al., 2019).

---

## [Decision Letter · Decision Letter 1]

23 Nov 2020

PONE-D-20-15837R1

Impact of gestational low-protein intake on embryonic kidney microRNA expression and in the nephron progenitor cells of the male offspring fetus

PLOS ONE

Dear Dr. Gontijo,

Thank you for submitting your manuscript to PLOS ONE. After careful consideration, we feel that it has merit but does not fully meet PLOS ONE’s publication criteria as it currently stands. Therefore, we invite you to submit a revised version of the manuscript that addresses the points raised during the review process by reviewer 2.

We look forward to receiving your revised manuscript.

Kind regards,

Emmanuel A Burdmann

Academic Editor

PLOS ONE

Reviewers' comments:

Reviewer's Responses to Questions

**Comments to the Author**

1. If the authors have adequately addressed your comments raised in a previous round of review and you feel that this manuscript is now acceptable for publication, you may indicate that here to bypass the “Comments to the Author” section, enter your conflict of interest statement in the “Confidential to Editor” section, and submit your "Accept" recommendation.

Reviewer #2: (No Response)

2. Is the manuscript technically sound, and do the data support the conclusions?

Reviewer #2: Yes

3. Has the statistical analysis been performed appropriately and rigorously? 

Reviewer #2: Yes

4. Have the authors made all data underlying the findings in their manuscript fully available?

Reviewer #2: Yes

5. Is the manuscript presented in an intelligible fashion and written in standard English?

Reviewer #2: Yes

6. Review Comments to the Author

Reviewer #2: The authors have significantly improved the manuscript, both scientifically and language-wise, however a number of flaws remain to be addressed. Specific comments follow below:

1. The authors were not as careful as they should be in their reply, copying and pasting their first response to reviewer 1 also as their first response to reviewer 2, including the term ”reviewer 1”.

2. It is unclear which experiments were redone; please specify them.

3. It is unclear which experiments had their numbers increased; please specify them.

4. It is true that the language quality improved, however the amount of sentence structure/clarity problems and grammatical mistakes remains above an acceptable level, both in the manuscript and in the reply to the reviewers. Please have the manuscript go through a native English speaker.

5. The authors have appropriately addressed my previous major comments 1 and 2, however such a point is unclear in the current manuscript. Please add this information to the revised version of the manuscript.

6. The modified versions of sentences 3.1 and 3.7 (re: previous major comment 3) are understandable but remain inappropriately written.

7. As a follow-up to my previous major comment 4, the authors state that they are currently performing studies to investigate autophagy and apoptosis pathways in nephron progenitor cells of metanephric mesenchyme during renal development, in the offspring of mice submitted to gestational protein restriction. Please include such data, at least part of them, in the current manuscript.

8. The authors have addressed my previous major comments 7, 11 and 12, however there are significant English problems in the corresponding texts.

9. My previous major comment 10 was not properly addressed, since the authors’ statement “the current study established that the let-7 family of miRNAs promotes MYC expression through transcriptionally induced let-7 repressor, LIN28 enhancement and posttranscriptional expressed LIN28 RNA binding-protein, promoting downregulation upon LP kidney cells differentiation” would require generation of data on Lin28 expression.

7. PLOS authors have the option to publish the peer review history of their article (what does this mean?). If published, this will include your full peer review and any attached files.

Reviewer #2: No

---

## [Author Response · Author response to Decision Letter 1]

8 Dec 2020

Response to Reviewers

PONE-D-20-15837

 The gestational low-protein intake impact in microRNA expression of the kidney progenitor cells in male offspring fetuses

Dear Editor:

We are attaching the re-revised version and response to the Reviewers of the Manuscript PONE-D-20-15837. The gestational low-protein intake impact in microRNA expression of the kidney progenitor cells in male offspring fetuses submitted by Sene et al. resubmitted as a full paper. We want to beware of the errors during the responses to the pertinent comments emanating by Reviewer 2 during the first evaluation of this manuscript. There is no justification for the carelessness that occurred. However, all occurred without bad faith and can be partially justified by the inexperience of the Author and the inattentive reading of the senior Author, who at the same time answered other criticisms from reviewers of other manuscripts sent for publication. I have read and have abided by the statement of ethical standards for documents submitted to the Plos One and the additional information that all authors have approved the final article. This manuscript has not been submitted or published in any other journal. At first, thank you very much for the reviewer's comments, suggestions, and criticisms. Practically all sections of the manuscript were rewritten entirely, experiments are redone, the number of experiments has been increased, and many reviewer suggestions were now included in that new version. The grammatical and typographical errors have been corrected. The Introduction, Material, and Method and Discussion sections of the manuscript were revised and completely rewritten to include reviewers' suggestions and comments.

Hopefully, this edited manuscript version could now be better considered for publication by this prestigious Journal.

Sincerely yours,

Patricia Aline Boer, Ph.D

Internal Medicine Department, School of Medicine, 

Campinas State University, Campinas, SP, Brazil. 

Reviewer 2 comments to the Author

Reviewer #2: The authors have significantly improved the manuscript, both scientifically and language-wise. However, several flaws remain to be addressed. Specific comments follow below:

 1. The authors were not as careful as they should be in their reply, copying and pasting their first response to reviewer 1 as their first response to reviewer 2, including the term ”reviewer 1”.

 2. It is unclear which experiments were redone; please specify them.

 3. It is unclear which experiments had their numbers increased; please specify them.

Response from the authors:

REF.: PONE-D-20-15837

R.: 1, 2, and 3 questions. 

We want to thank Reviewer 2 very much for the spending time and careful reading and to beware of the errors during the responses to the pertinent comments emanating during the first evaluation of this manuscript. There is no justification for the carelessness that occurred. However, all occurred without bad faith and can be partially justified by the inexperience of the Author and the inattentive reading of the senior Author, who at the same time answered other criticisms from reviewers of other manuscripts sent for publication. As suggested, the document was submitted for revision by a native English speaker. Among the errors that occurred during the response to Reviewer 2, the authors stated that they remade and carried out new experiments. This gross error for which we apologize was due, as stated above, to the simultaneous response of papers being reviewed simultaneously. Embarrassed, we apologize again. We have much appreciated your comments and suggestions. Practically, all manuscript sections were entirely rewritten, and many reviewer suggestions were included in that new version. The Introduction, Material, and Method and Discussion sections of the manuscript were revised and completely rewritten to include the suggestions and comments of reviewer 2.

 4. It is true that the language quality improved. However, the amount of sentence structure/clarity problems and grammatical mistakes remains above an acceptable level, both in the manuscript and in reply to the reviewers. Please have the manuscript go through a native English speaker.

R.: 4 comment

As suggested, the document was submitted for revision by a native English speaker.

 5. The authors have appropriately addressed my previous major comments 1 and 2, however such a point is unclear in the current manuscript. Please add this information to the revised version of the manuscript.

Response to comment 5: The changes were included in Figures 1 and 2 legends to read:

Figure 1. Expression of miRNAs in the metanephros from the 17th day LP fetus compared to their expression level in the control group. Reference genes U6 and U87, protein complexes composed of small nuclear RNAs (snRNAs), were used to normalize each miRNA expression. The authors established a cutoff point variation of 1.3 (upwards) or 0.65 (downwards), and data are expressed as fold change (mean ± SD, n = 4) concerning the control group. * p≤0.05: statistical significance versus NP. 

Figure 2. Expression of mRNA estimated by SyBR green RT-qPCR of metanephros from the 17th day LP fetus. The expression was normalized with GAPDH. The authors established a cutoff point variation of 1.3 (upwards) or 0.65 (downwards), and data are expressed as fold change (mean ± SD, n = 4) concerning the control group. * p≤0.05: statistical significance versus NP. 

 6. The modified versions of sentences 3.1 and 3.7 (re: previous major comment 3) are understandable but remain inappropriately written.

R.: 6 comment

Sentence 3.1 was changed to read:

Authors demonstrated that gestational low-protein (LP) intake offspring presents a lower birth weight, reduced nephron numbers and renal salt excretion, arterial hypertension, and renal failure development compared to regular protein (NP) intake rats in adulthood. The current study evaluated the different miRNAs and kidney predicted target gene expression in gestational 17-days LP (DG-17) fetal metanephros seeking to elucidate some of the molecular pathways involved in the proliferation and differentiation of renal embryonic/fetal cell.

Sentence 3.7 was changed to read:

By the way, the exhaustion of the remaining CM progenitor cell endowment, in turn, predispose reduced nephron numbers, arterial hypertension development, and renal disorders in adult age (Figure 10). In the current study, confirming Boivin et al. (2015), we may state that increased CM β-catenin might disrupt UB ramifications and nephrogenesis. Have been previously known that growth factor glial-derived neurotrophic factor (GDNF) genes encoding, a crucial regulator of UB outgrowth, acts via c-Ret tyrosine kinase receptor and Gfra1 co-receptor (Brodbeck et al., 2004; Davis et al., 2015). In the 17-DG LP offspring, a significant increase in c-Ret receptor coding mRNA would theoretically lead to a rise in UB ramifications. However, in the present study, the GDNF expression was unchanged, being plausible to suppose that, despite the increase of c-Ret mRNA, the UB branching was reduced. Prior data observed in our Lab, showed a reduction of 28.3% in ureteric bud branches after 14.5 days of gestational protein restriction (Mesquita et al., 2010b) could be associate with a 28% reduction in labeling for Six2 cells in the MM, despite unchanged GDNF transcription.

 7. As a follow-up to my previous major comment 4, the authors state that they are currently performing studies to investigate autophagy and apoptosis pathways in nephron progenitor cells of metanephric mesenchyme during renal development, in the offspring of mice submitted to gestational protein restriction. Please include such data, at least part of them, in the current manuscript.

R.: 7 comment

Studies are being done in a doctorate thesis in our Lab using mice from the C57BL / 6-TgCAG-RFP / EGFP / Map1lc3b1Hill / J strain, transgenic autophagy. However, the studies are ongoing, and only preliminary results have been obtained.

 8. The authors have addressed my previous major comments 7, 11, and 12, however there are significant English problems in the corresponding texts.

R.: 8 comment

The grammatical and typographical errors have been edited.

 9. My previous major comment 10 was not properly addressed, since the authors’ statement “the current study established that the let-7 family of miRNAs promotes MYC expression through transcriptionally induced let-7 repressor, LIN28 enhancement and posttranscriptional expressed LIN28 RNA binding-protein, promoting downregulation upon LP kidney cells differentiation” would require generation of data on Lin28 expression.

R.: 9 comment

The question raised by the reviewer is pertinent. However, we do not have enough tissue from all the animals studied to perform the analyzes. In this way, we try not to be affirmative in the discussion of the work, just carefully suggesting the possibility that, at least in part, the suppressive effects of the expression of let-7 miRNAs are indirectly due to a reduction in lin28. As we answered above, we are carrying out complementary studies on the routes shown here, so we hope in the short term to have an answer to this question.

Thus, include the text below to read:

On the other hand, Yermalovich et al. (2019) have demonstrated that the overexpression of Lin28b, an RNA-binding protein, is associated with suppressive let-7 miRNAs. While the lin28 and let-7 genes are well-established regulators of ontogenic timing in invertebrates, the role of these in mammalian organ development was not fully understood. In the current study, the enhanced let-7a-5p miRNA expression in the LP fetus could be associated with reducing CM cell proliferation, compromising the whole nephrogenesis relative to the NP group. In this way, we may hypothesize the unprecedented findings of CM cell proliferation suppression and early cessation of nephrogenesis caused by increased let-7 miRNAs may occur directly or indirectly via the transient reduced expression of Lin28b in the 17-DG LP.

---

## [Decision Letter · Decision Letter 2]

21 Dec 2020

PONE-D-20-15837R2

Impact of gestational low-protein intake on embryonic kidney microRNA expression and in the nephron progenitor cells of the male offspring fetus

PLOS ONE

Dear Dr. Gontijo,

Thank you for submitting your manuscript to PLOS ONE. After careful consideration, we feel that it has merit but does not fully meet PLOS ONE’s publication criteria as it currently stands. Therefore, we invite you to submit a revised version of the manuscript that addresses the minor points raised during the review process.

We look forward to receiving your revised manuscript.

Kind regards,

Emmanuel A Burdmann

Academic Editor

PLOS ONE

Reviewers' comments:

Reviewer's Responses to Questions

**Comments to the Author**

1. If the authors have adequately addressed your comments raised in a previous round of review and you feel that this manuscript is now acceptable for publication, you may indicate that here to bypass the “Comments to the Author” section, enter your conflict of interest statement in the “Confidential to Editor” section, and submit your "Accept" recommendation.

Reviewer #2: (No Response)

2. Is the manuscript technically sound, and do the data support the conclusions?

Reviewer #2: Yes

3. Has the statistical analysis been performed appropriately and rigorously? 

Reviewer #2: Yes

4. Have the authors made all data underlying the findings in their manuscript fully available?

Reviewer #2: Yes

5. Is the manuscript presented in an intelligible fashion and written in standard English?

Reviewer #2: Yes

6. Review Comments to the Author

Reviewer #2: My comments have been adequately addressed or justified, however two points remain to be fulfilled/considered.

1. The language quality gave one more step ahead, however it still needs improvement to reach an appropriate level. Incomplete sentences, inappropriately structured sentences and grammatical errors are still frequent, particularly in the discussion. At this point, therefore, my recommendation is to have an editing service review the manuscript.

2. The authors are presently performing studies to investigate autophagy and apoptosis pathways in nephron progenitor cells of metanephric mesenchyme during renal development in the offspring of mice submitted to gestational protein restriction. Initial data have been generated. Given the relevance of this issue to the present study and the additional time from the last revision, do the current results include any significantly robust piece of data that could expand this aspect in the present manuscript?

7. PLOS authors have the option to publish the peer review history of their article (what does this mean?). If published, this will include your full peer review and any attached files.

Reviewer #2: No

---

## [Author Response · Author response to Decision Letter 2]

3 Jan 2021

Response to Reviewers

PONE-D-20-15837

 The gestational low-protein intake impact in microRNA expression of the kidney progenitor cells in male offspring fetuses

Dear Editor:

We are attaching the Edited version and response to the Reviewers of the Manuscript PONE-D-20-15837. The gestational low-protein intake impact in microRNA expression of the kidney progenitor cells in male offspring fetuses submitted by Sene et al. resubmitted as a full paper. Practically, all manuscript sections were rewritten entirely, and many reviewer suggestions were now included in that new version. The grammatical and typographical errors have been corrected, and the manuscript was submitted to the editing service to review. The Introduction, Material, and Method and Discussion sections of the document were revised and completely rewritten to include reviewers' suggestions and comments.

Hopefully, this edited manuscript version could now be better considered for publication by this prestigious Journal. 

Sincerely yours,

Patricia Aline Boer, Ph.D

Internal Medicine Department, School of Medicine, 

Campinas State University, Campinas, SP, Brazil. 

Reviewer 2 comments to the Author

Reviewer #2: My comments have been adequately addressed or justified, however two points remain to be fulfilled/considered.

1. The language quality gave one more step ahead, however it still needs improvement to reach an appropriate level. Incomplete sentences, inappropriately structured sentences and grammatical errors are still frequent, particularly in the discussion. At this point, therefore, my recommendation is to have an editing service review the manuscript.

Response from Authors

R.1: Practically, all manuscript sections were rewritten entirely, and many reviewer suggestions were now included in that new version. The grammatical and typographical errors have been corrected, and the manuscript was submitted to the editing service to review. The Introduction, Material, and Method and Discussion sections of the document were revised and completely rewritten to include prior reviewers' suggestions and comments.

2. The authors are presently performing studies to investigate autophagy and apoptosis pathways in nephron progenitor cells of metanephric mesenchyme during renal development in the offspring of mice submitted to gestational protein restriction. Initial data have been generated. Given the relevance of this issue to the present study and the additional time from the last revision, do the current results include any significantly robust piece of data that could expand this aspect in the present manuscript?

R2: Dear reviewer, the authors choose to present the results on autophagy and apoptosis in a future manuscript for two main reasons: 1. it is part of a doctoral study by another student, which includes a series of other evaluations, which at this moment only presents preliminary results; 2. Studies on this topic have suffered a significant delay since the University suspended on-site activities, practically throughout the year 2020, which caused a considerable delay in obtaining the results. This delay also includes the breeding and genetic control of mice from the C57BL / 6-TgCAG-RFP / EGFP / Map1lc3b1Hill / J strain, essential for reasonable experimental control. However, the studies are ongoing, and preliminary results have been obtained.

---

## [Decision Letter · Decision Letter 3]

18 Jan 2021

Impact of gestational low-protein intake on embryonic kidney microRNA expression and in nephron progenitor cells of the male fetus

PONE-D-20-15837R3

Dear Dr. Gontijo,

We’re pleased to inform you that your manuscript has been judged scientifically suitable for publication and will be formally accepted for publication once it meets all outstanding technical requirements.

Kind regards,

Emmanuel A Burdmann

Section Editor

PLOS ONE

Additional Editor Comments (optional):

Reviewers' comments:

Reviewer's Responses to Questions

**Comments to the Author**

1. If the authors have adequately addressed your comments raised in a previous round of review and you feel that this manuscript is now acceptable for publication, you may indicate that here to bypass the “Comments to the Author” section, enter your conflict of interest statement in the “Confidential to Editor” section, and submit your "Accept" recommendation.

Reviewer #2: All comments have been addressed

2. Is the manuscript technically sound, and do the data support the conclusions?

Reviewer #2: (No Response)

3. Has the statistical analysis been performed appropriately and rigorously? 

Reviewer #2: (No Response)

4. Have the authors made all data underlying the findings in their manuscript fully available?

Reviewer #2: (No Response)

5. Is the manuscript presented in an intelligible fashion and written in standard English?

Reviewer #2: (No Response)

6. Review Comments to the Author

Reviewer #2: (No Response)

7. PLOS authors have the option to publish the peer review history of their article (what does this mean?). If published, this will include your full peer review and any attached files.

Reviewer #2: No

---

## [Editor Report · Acceptance letter]

27 Jan 2021

PONE-D-20-15837R3 

Impact of gestational low-protein intake on embryonic kidney microRNA expression and in nephron progenitor cells of the male fetus 

Dear Dr. Gontijo:

I'm pleased to inform you that your manuscript has been deemed suitable for publication in PLOS ONE. Congratulations! Your manuscript is now with our production department. 

Kind regards, 

on behalf of

Dr. Emmanuel A Burdmann 

Section Editor

PLOS ONE